# Scenario Diffusion: Controllable Driving Scenario Generation With Diffusion

**Ethan Pronovost**[*]   **Meghana Reddy Ganesina**   **Noureldin Hendy**   **Zeyu Wang**[†]
**Andres Morales**   **Kai Wang**   **Nicholas Roy**
Zoox
{epronovost,mrganesina,nhendy,zewang,andres,kai,nroy}@zoox.com

## Abstract

Automated creation of synthetic traffic scenarios is a key part of validating the safety of autonomous vehicles (AVs). In this paper, we propose Scenario Diffusion, a novel diffusion-based architecture for generating traffic scenarios that enables *controllable* scenario generation. We combine latent diffusion, object detection and trajectory regression to generate distributions of synthetic agent poses, orientations and trajectories simultaneously. To provide additional control over the generated scenario, this distribution is conditioned on a map and sets of tokens describing the desired scenario. We show that our approach has sufficient expressive capacity to model diverse traffic patterns and generalizes to different geographical regions.

## 1   Introduction

Automated creation of synthetic traffic scenarios is a key part of validating the safety of autonomous vehicles (AVs). To efficiently target rare and safety critical scenarios we would like to direct scenario generation to produce specific types of events. Prior methods for heuristically created scenarios tend to be of limited complexity and miss a large number of possible real-world situations [5, 7, 16, 17]. Recent works using deep learning models [2, 6, 32] are able to produce complex scenarios conditioned on a map region, but do not offer additional controls over the generation process. In this paper we propose Scenario Diffusion: a novel architecture for generating realistic traffic scenarios that enables *controllable* scenario generation.

Our problem setting is to generate a set of bounding boxes with associated trajectories that describe the location and behavior of agents in a driving scenario. We accomplish this scenario generation using a denoising diffusion generative model [12, 27, 29]. We condition our model on both a map image and a set of tokens describing the scenario. We leverage these tokens to provide a variable rate of control over the generated scenario, so that the diffusion model can generalize to complex scenes conditioned only on a small set of tokens that control both a subset of the individual agents and global scene properties. Further, this variable rate of control allows us to learn the model from partially tokenized training instances, substantially reducing the labelling requirements.

Motivated by the insight that the instantaneous position of each agent is inextricably linked to their behaviors, we combine latent diffusion [24], object detection, and trajectory regression to simultaneously generate both oriented bounding boxes and trajectories, providing a generative model of both the static placement of agents and their behaviors. We evaluate Scenario Diffusion at generating driving scenarios conditioned on only the map, and with additional conditioning tokens as well. Finally, we provide an analysis of the generalization capabilities of our model across geographical regions, showing that our diffusion-based approach has sufficient expressive capacity to model diverse traffic patterns.

---

[*]Corresponding author
[†]Work done during an internship at Zoox

37th Conference on Neural Information Processing Systems (NeurIPS 2023).

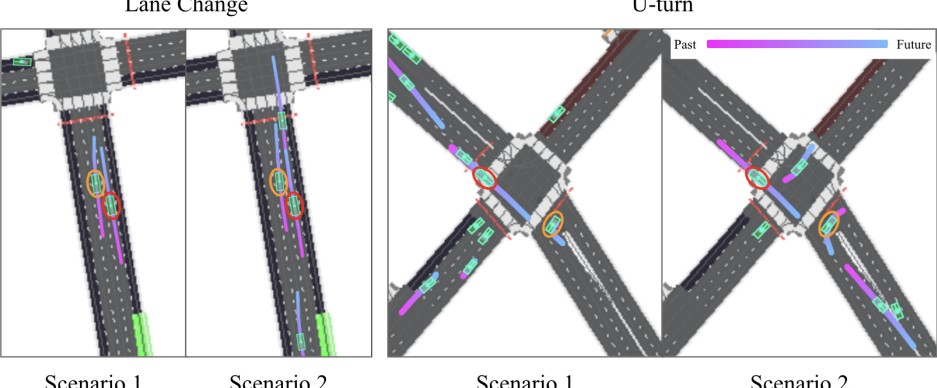

Figure 1: Driving scenarios generated using the map location and token features describing the two circled vehicles. Vehicle bounding boxes are shown in green with their trajectories colored from pink in the past to blue in the future. The vehicle circled in red represents the AV, and the vehicle circled in orange is a key interacting vehicle. The token features describe a desired position, heading, size, speed, and trajectory curvature of the two circled agents. Using this description, the model generates scenarios that include two such agents and adds additional agents in a realistic manner to fill the scenario. The two scenarios in each example are generated using the same conditional inputs and different initial diffusion noise samples. By using tokens, we do not need to fully define the circled vehicles' trajectories, and the model produces different possible scenarios that conform to the abstract description encoded by the token.

## 2 Problem Setting

We aim to generate traffic scenarios conditioned on a map and an optional description of the scenario. Similar to prior works [2, 6, 32] we represent traffic scenarios abstractly with oriented bounding boxes and trajectories[3]. AV software is typically divided into *perception* and *motion planning*. The interface between these components describes agents with bounding boxes, historical trajectories, and additional attributes. In simulation this abstract representation is synthesized to evaluate the motion planning component in isolation, alleviating the need for expensive photorealistic simulation.

Fig. 1 demonstrates the type of controllability we aim to achieve. We condition the generative model by describing the state of two vehicles: the AV (circled in red) and one key interacting agent (circled in orange). The model is able to generate a diverse set of scenarios that include these two agents and additional ones. Notably, describing the two key agents does not require having observed such agents in the real world (e.g. by copying these two agents from an existing log). Instead, the model is able to generalize to generating novel agents based on the given description.

### 2.1 Scenario Representation

In this work we use topdown birds' eye view (BEV) representations similar to [1, 2, 13, 32] for the inputs to convolutional networks and sparse representations of oriented boxes and trajectories as the final output of the decoder.

**Topdown Representations** The map context is represented as a multi-channel BEV image $m$ of shape $C_m \times H \times W$ that contains information about road geometry, regions of interest (e.g. driveways, crosswalks), and traffic control devices (e.g. stop signs, traffic lights). Agents are also represented in a multi-channel BEV image $x$ of shape $C_x \times H \times W$ that encodes each agent's bounding box, heading, and trajectory. Sec. 4.1 describes two different datasets with different rendering styles for $m$ and $x$ that both yield good results with our model.

---

[3]Our simulations are abstract representations that are produced by perception systems. While photorealistic simulation would also allow the perception system to be tested in the loop, in this work we focus on scenario generation that tests planning and decision making for computational tractability.

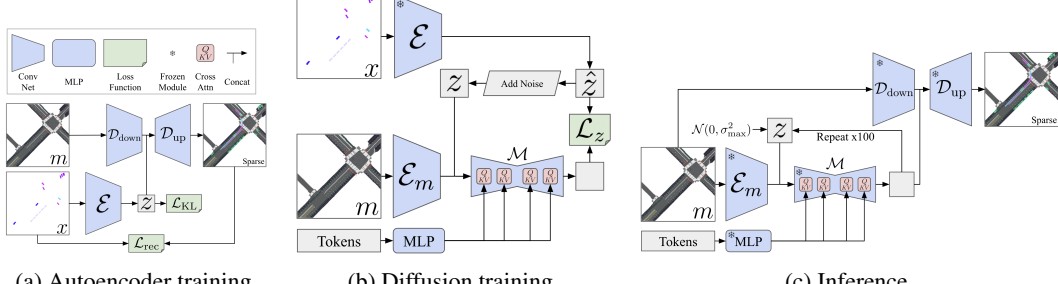

(a) Autoencoder training.    (b) Diffusion training.    (c) Inference.

Figure 2: Model architectures for training and inference. (a) The autoencoder takes as input birds' eye view renderings of the map and entities and outputs sparse bounding box detections with trajectories. (b) The diffusion model is trained to denoise latent embeddings from the autoencoder using a birds' eye view rendering of the map and tokens describing the scene. (c) To perform inference, initial random noise is iteratively denoised and then decoded to generate bounding boxes and trajectories.

**Sparse Representations** We also represent the agents in the scene as a set of oriented bounding boxes, with associated trajectories included as an additional feature vector per box. The oriented bounding box is defined by a center position, heading, length, and width and the agent trajectory as a sequence of future and past poses spaced at uniform intervals in time, as is common in motion forecasting [1, 8, 22, 23, 25, 31, 33, 38, 40], with the bounding box pose representing the "current timestep". This sparse representation is used as the labels for the autoencoder's detection task and to compute tokens describing agents in the scenario.

# 3 Method

As in [24], our diffusion model consists of two parts: an autoencoder to transform between the data space and a latent space and a diffusion model that operates in this latent space.

## 3.1 Scenario Autoencoder

The scene autoencoder is a variational autoencoder (VAE) [11] that learns to encode and decode sets of agents as shown in Fig. 2a. The VAE is trained with a combination of a reconstruction loss $\mathcal{L}_{\text{rec}}$ on the decoder output and a KL divergence loss $\mathcal{L}_{\text{KL}}$ on the latent embedding. The architecture is based on the autoencoder of [24], but we train the model to perform anchor-free one-to-one detection instead of pixelwise image reconstruction.

An encoder $\mathcal{E}$ takes the BEV image of the agents in the scene $x$ and outputs a latent embedding $z = \mathcal{E}(x)$ of shape $C_z \times H' \times W'$, where $H' = H/2^f$ and $W' = W/2^f$ for a downsampling factor $f \in \mathbb{N}$. Since the encoder is designed to capture the placement of agents, there was empirically no benefit to providing the map image as input to the encoder. This latent embedding is given to the decoder $\mathcal{D}$ along with the BEV map image $m$ to obtain the detection output $y = \mathcal{D}(z, m)$ of shape $C_y \times H'' \times W''$. Because we output detections, $H''$ and $W''$ do not need to equal $H$ and $W$, the original input shapes. In practice, having a lower output resolution provides significant memory savings during training. The decoder downsamples the map image $m$, concatenates the latent embedding $z$, and then upsamples to produce the final output: $\mathcal{D}(z, m) = \mathcal{D}_{\text{up}}(z, \mathcal{D}_{\text{down}}(m))$.

**Detection Outputs** The decoder output $y$ represents oriented bounding box detections and trajectories. Each pixel of $y$ represents one box proposal, with different channels representing the probability, position, heading, length, width, and trajectory of the box. Concretely, the first channel is a sigmoid logit for the box probability. The second and third channels represent the cosine and sine of the box orientation. The next four channels represent the distances from the pixel center to the front, left, back, and right edges of the box in log scale. These seven channels define the oriented bounding box for the agent.

The trajectory describes the agent's pose from $T$ seconds in the past to $T$ seconds in the future at one second intervals. For each timestep $t \in \{1, \ldots, T\}$ of the future trajectory three channels of $y$ represent the delta between the pose at $t$ and the pose at $t-1$ (with channels for x, y, and heading). These deltas are accumulated to produce the final trajectory starting from the center pose of the

oriented bounding box. The past trajectory is handled symmetrically (i.e. deltas between $t$ and $t + 1$ for $t \in \{-T, \ldots, -1\}$). In total 1 channel encodes the box probability, 6 channels encode the oriented bounding box, and $6T$ channels encode the trajectory.

**Reconstruction Loss**  The reconstruction loss used to train the autoencoder adapts the one-to-one detection loss of [30] to accommodate oriented boxes. Given the simplicity of our detection problem (perfectly rendered rectangles with disambiguated orientations) we find that replacing the IoU loss (which is non-differentiable with oriented boxes) with an L2 loss on the box vertices yields detection performance sufficient for training the diffusion model. Following [30], each ground truth box is matched with one predicted box by selecting the predicted box that minimizes a matching score between it and the ground truth box. This matching score is a weighted combination of a classification score, L1 loss on the box parameters, and an L2 loss on the box vertices. A classification loss encourages matched predicted boxes to have high probability and unmatched predicted boxes to have low probability, and regression losses encourage matched predicted boxes and trajectories to equal their corresponding ground truth box. We use a weighted combination of an L1 loss on box parameters, L2 loss on box vertices, L2 loss on trajectory positions, and cosine loss on the trajectory headings. Further details on the reconstruction loss can be found in Appendix B.2.

## 3.2 Diffusion Fundamentals

Once the autoencoder is trained, a diffusion model is trained in the latent space of the autoencoder. This means that the diffusion model does not operate on bounding boxes and trajectories directly. We use the EDM diffusion algorithm [18] to train our model and generate novel scenarios.

**Training**  Let $\hat{z} = \mathcal{E}(x)$ be the latent embedding obtained from a trained and frozen encoder. Each example can also include associated conditional information $c$ (e.g., map information, tokens). We discuss options for this conditional data in following sections.

For each training example we sample $\sigma$ according to a log-normal distribution (i.e. $\log \sigma \sim \mathcal{N}\left(P_\mu, P_\sigma^2\right)$ with hyperparameters $P_\mu$ and $P_\sigma$). We then create a noisy sample $z = \hat{z} + \sigma\epsilon$, where $\epsilon$ has the same shape as $\hat{z}$ and is sampled from the standard normal distribution.

We train the denoising model $\mathcal{M}(z; c, \sigma)$ by minimizing the reconstruction loss

$$\mathcal{L} = \lambda(\sigma) \cdot c_{\text{out}}(\sigma)^2 \left\| \mathcal{M}(c_{\text{in}}(\sigma) \cdot z; c, \sigma) - \frac{\hat{z} - c_{\text{skip}}(\sigma) \cdot z}{c_{\text{out}}(\sigma)} \right\|_2^2 \tag{1}$$

where $c_{\text{in}}(\sigma)$, $c_{\text{out}}(\sigma)$, $c_{\text{skip}}(\sigma)$, and $\lambda(\sigma)$ are all scalar functions of $\sigma$ as defined in [18]. In practice this model $\mathcal{M}$ is a conditional Unet as in [3, 12, 28]. This process is depicted in Fig. 2b. Given a noisy sample $z$, the denoised version $\hat{z}$ can be estimated as

$$M(z; c, \sigma) = c_{\text{skip}}(\sigma) \cdot z + c_{\text{out}}(\sigma) \cdot \mathcal{M}(c_{\text{in}}(\sigma) \cdot z; c, \sigma) \tag{2}$$

**Inference**  To generate novel samples during inference, we start with an initial noisy sample $z \sim \mathcal{N}\left(0, \sigma_{\text{max}}^2 \mathbf{I}\right)$. This sample is iteratively refined according to the reverse process ODE as described in [18]. The gradient of the log probability distribution can be approximated as $\nabla_z \log p(z; c, \sigma) \approx (M(z; c, \sigma) - z)/\sigma^2$. This simplifies the reverse process ODE to

$$\frac{\mathrm{d}z}{\mathrm{d}\sigma} = -\sigma \nabla_z \log p(z; c, \sigma) \approx \frac{z - M(z; c, \sigma)}{\sigma} \tag{3}$$

After integrating the sample $z$ from $\sigma = \sigma_{\text{max}}$ to $\sigma = 0$ using Euler integration, it is passed through the decoder to obtain the generated boxes and trajectories $\mathcal{D}(z, m)$. This procedure is depicted in Fig. 2c. During inference we keep the generated boxes with probability above a fixed threshold chosen by optimizing a metric that compares generated and ground truth boxes (described in Sec. 4.1). We then filter any overlapping boxes, keeping the highest probability box of those that overlap.

## 3.3 Diffusion Conditioning

Conditional Unet models have two primary ways to provide the conditional information $c$: concatenating to the noisy sample that is input to the denoising model and applying cross attention over a set of tokens inside the denoising model. We explore both options in this work.

### 3.3.1 Map Conditioning

One method to include conditional data in the denoising model is via a conditional image that gets concatenated with the noisy sample input to the denoising model [24]. In this work we use the BEV map image $m$ as described in Sec. 2.1 as the conditional image. The map image $m$ is first processed by a separate encoder $\mathcal{E}_m$ to downsample it to the same pixel dimension as $z$. The encoded map $\mathcal{E}_m(m)$ is concatenated with $z$ in the channel dimension before being input to the denoising Unet $\mathcal{M}$.

### 3.3.2 Token Conditioning

We can also condition diffusion models on a set of tokens via a cross-attention mechanism [34] within the denoising model [24]. In image generation these tokens often come from text embeddings of a caption for the image [26]. While we do not have datasets of driving scenarios fully annotated with text captions, we can still use the concept of conditional tokens that describe aspects of the scenario. It is exactly these tokens that provide *controllability* over scenario generation. In this work we explore two types of tokens: *agent tokens* and *global scene tokens*.

**Agent Tokens**    Agent tokens are tokens that describe a single agent. In this work we use procedurally generated tokens (i.e. tokens that can be computed from available data in the dataset), so are able to provide a token for every agent. However that is not a requirement for this approach; if using human-labeled tokens it would be possible to train a diffusion model while only having partially labeled scenarios. The token feature vector for each agent expresses various quantities that we might want associated with that agent, such as the agent's position, speed, length, or motion profile. In this work the meaning of the token feature vector is fixed for a given experiment. One potential direction for future work is to have a single model support multiple types of agent token feature embeddings (e.g. to allow the user to specify the length for one agent and the speed for another).

To enable the model to generate agents not described by tokens we use a partial tokenization strategy. During training we achieve this by sampling a token mask probability $p_{\text{mask}}$ and masking out agent tokens with this probability. More information on this can be found in Sec. 4.3.1.

**Global Scene Tokens**    Global scene tokens encode scalar quantities that allow us to control global properties of the entire scenario. In this work we consider a single global scene token to represent $p_{\text{mask}}$, which encodes information about the number of agents to be generated relative to the number of tokens provided. During inference the value represented by this global scene token can be modulated to control the number of agents in the generated scenarios. By providing no agent tokens and setting $p_{\text{mask}} = 1$ in the global scene token we are able to generate scenarios conditioned on only the map from the same model.

**Processing**    Each type of token is processed by a separate MLP to get final embeddings in a common latent space, to be used as keys and values during cross-attention inside the denoising model. At different levels of the Unet each pixel from the intermediate image representation is used as a query, and the agent and global scene tokens are used as the keys and values. This architecture can support any number of tokens. We refer the reader to [24] for more details on the cross-attention mechanism.

## 4 Experiments

### 4.1 Experimental Setup

**Datasets**    We use two datasets in this work. The Argoverse 2 motion forecasting dataset [36] contains 250,000 driving scenarios over 6 geographical regions representing 763 hours of total driving. We use the provided training and validation splits. More information about this dataset can be found in Appendix A.1.

The second dataset is an internal dataset containing 6 million real world driving scenarios from Las Vegas (LV), Seattle (SEA), San Francisco (SF), and the campus of the Stanford Linear Accelerator Center (SLAC). Training and validation data are not separated by location, so a given validation example may have training examples from the same location at other times. More information about this dataset can be found in Appendix A.2.

Table 1: Quality metrics for generated scenarios on Argoverse.

| Method | MMD² Positions (↓) | MMD² Headings (↓) | MMD² Velocity (↓) | Traj On Drivable | Lane Heading Difference |
|---|---|---|---|---|---|
| Ground Truth Log | - | - | - | 0.900 (0.000) | 0.203 (0.000) |
| Random Log Selection | 0.142 (0.001) | 0.386 (0.001) | 0.104 (0.001) | 0.407 (0.001) | 1.497 (0.004) |
| TrafficGen [6] Placement Model | 0.113 (0.002) | 0.124 (0.003) | **0.054** (0.002) | - | - |
| Scenario Diffusion (No Map) | 0.142 (0.006) | 0.352 (0.012) | 0.091 (0.007) | 0.411 (0.009) | 1.504 (0.004) |
| Scenario Diffusion | **0.093** (0.006) | **0.108** (0.007) | **0.055** (0.003) | 0.895 (0.008) | 0.271 (0.017) |

For both datasets, the scenes are centered on the autonomous vehicle. As the autonomous vehicle is treated as just another vehicle during training, the model almost always places a vehicle in the center of the scene during inference. We use a 2 second time horizon for both the future and the past trajectory (i.e. the bounding box represents $t = 0$ and the trajectory goes from $t = -2$ to $t = 2$). This time window is sufficient to populate the agent history for many motion forecasting methods [1, 8, 22, 23, 25, 31, 33, 38, 40].

**Training** See Appendix B and for details of training the autoencoder and Appendix C for details of training the diffusion model.

**Inference** We use 80% as the threshold for the generated box probabilities. See Appendix D.2 for the ablation experiment used to select this threshold. We note that the performance of Scenario Diffusion is relatively robust to choices of probability threshold between 50% and 90%.

**Metrics** We consider metrics to compare the generated and data distributions and measure the quality of generated scenarios. One popular metric for comparing two distributions using samples is the maximum mean discrepancy (MMD) [9]. While this metric is typically defined in terms of two distributions, in practice we compute this metric using samples drawn from the two distributions. Given two sets $A$ and $B$ generated by sampling from two distributions and some kernel $k$, the maximum mean discrepancy is defined as

$$\text{MMD}^2(A, B) = \frac{1}{|A|^2} \sum_{a_1, a_2 \in A} k(a_1, a_2) + \frac{1}{|B|^2} \sum_{b_1, b_2 \in B} k(b_1, b_2) - \frac{2}{|A||B|} \sum_{a \in A, b \in B} k(a, b) \quad (4)$$

Similar to [6, 32], we use a Gaussian kernel $k$ and apply this metric to the sets of agent center positions, heading unit vectors, and velocities, all in $\mathbb{R}^2$.

To measure the quality of generated trajectories we compute the fraction of waypoints that fall within the drivable area, averaged over all generated agents and all 5 trajectory waypoints. For each predicted pose along the trajectory we also compute the minimum angle difference (in radians) between the pose heading and the heading of all lanes at that location. These metrics do not have a clear optimal value (trajectories in the Argoverse dataset sometimes leave the drivable area and aren't perfectly aligned with the lane tangent); we compare the metric between the dataset and generated scenarios.

**Baselines** As a simple heuristic baseline we implement Random Log Selection, which takes the agents from a random sample in the training dataset (ignoring map information). We also report the trajectory metrics on the ground truth validation samples themselves.

To compare our method to previous autoregressive approaches we adapt TrafficGen [6] to the Argoverse 2 [36] dataset. See Appendix E for more details on the modifications needed.

### 4.2 Map-Conditioned Scenario Generation

We first evaluate Scenario Diffusion in generating scenarios conditioned on only map data, the problem setting explored in prior works [2, 6, 32]. In Tab. 1 we report MMD metrics that compare generated and ground truth scenes and trajectory metrics that measure how well the trajectories conform to the map. We evaluate Scenario Diffusion models with and without the map context $m$, and compare against the baselines described above.

Scenario Diffusion outperforms all other methods in the MMD metric for position and heading, and performs equally with TrafficGen in velocity. In the trajectory metrics, Scenario Diffusion produces similar values as those of the ground truth logs. As expected, Scenario Diffusion without map context performs similarly to random log selection, as both methods ignore the map context.

Global Scene Token = 0%        Global Scene Token = 50%        Global Scene Token = 90%

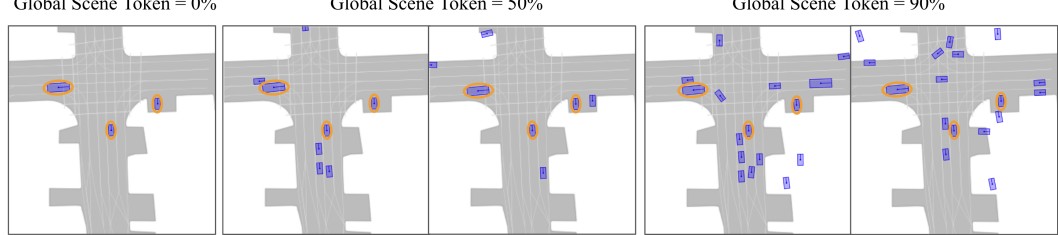

Figure 3: Generated Argoverse scenarios conditioned on the same map image and three agent tokens with different values for the global scene token. The generated agents corresponding to the three agent tokens are circled in orange. When the global scene token is set to non-zero values, the model reconstructs the three agents described by tokens and adds additional agents to fill in the scene. Trajectories are omitted from this figure for clarity.

Table 2: Agent token matching metrics on Argoverse

| Token Mask Probability $p_{mask}$ | 0% | 10% | 30% | 50% | 70% | 90% |
|---|---|---|---|---|---|---|
| Agent token match rate | 0.97 (0.01) | 0.97 (0.01) | 0.97 (0.01) | 0.97 (0.01) | 0.96 (0.01) | 0.96 (0.02) |
| Number of additional generated agents | 0.4 (0.3) | 3.0 (0.8) | 5.5 (0.9) | 8.0 (0.9) | 10.4 (0.8) | 12.6 (0.7) |

## 4.3  Token Conditioning

We next explore applications of token conditioning in Scenario Diffusion. Fig. 1 demonstrates how tokens can be used to generate scenarios with a specific type of interaction between the AV and another agent. Our approach is able to generate these interactions without having to fully specify the box and trajectory for the agent (e.g. by hand or by copying an agent from an existing log). By using more abstract features in the agent tokens we allow the model to generate different scenarios that meet the given description (e.g. we do not prescribe the specific trajectory of the u-turning agent).

The flexibility of tokens provides variable degrees of control over the generated scenarios. In this work we demonstrate models that use a variety of features in the agent tokens, such as position, heading, length, and speed. More details on the exact token features can be found in Appendix C.3.

### 4.3.1  Partial Tokenization

It is crucial that the learned model be robust to the presence or absence of user-provided tokens. To ensure this, during training we omit or "mask" supervision tokens so that the model learns to fill in additional agents in the scenario. During training we select $p_{mask}$ such that 40% of the time we keep all tokens (i.e. $p_{mask} = 0$) and the other 60% of the time we sample $p_{mask} \sim \text{Beta}(2, 1)$.[4] For each sample, the chosen masking probability is bucketed into one of ten bins ([0, 0.1), [0.1, 0.2), etc.) and embedded as a one-hot vector to define the global scene token. This global scene token and the agent tokens that remain after masking are used to condition the diffusion model as described in Sec. 3.3.2.

During inference we can modulate the value represented by the global scene token to control how many additional agents the model generates beyond those described by agent tokens. Fig. 3 shows scenarios generated by the model using a fixed map, three agent tokens, and the global scene token. All scenarios contain the three agents described by the tokens. As the value set in the global scene token increases, the model infers that there are additional agents not described by tokens. This approach allows a user to specify certain agents while having the model fill in the remainder of the scenario. Using this formulation we are able to emulate a model conditioned on only the map image by setting the global scene token to 100% and providing no additional agent tokens.

We first evaluate the model's ability to use the token information by matching generated agents to tokens. We take samples from the Argoverse validation dataset, compute agent tokens from the ground truth scene, and apply a fixed amount of agent token masking. We generate scenes conditioned on these tokens and compute a one-to-one matching between generated agents and agent tokens using the Hungarian algorithm [20]. We only consider pairs that are within 2.2 meters (we discretize position into bins of width 1.56m, and $1.56 * \sqrt{2} \approx 2.2$) and 0.2 radians. In Tab. 2 we report the

---

[4]Simpler choices such as sampling $p_{mask}$ from a uniform distribution over [0, 1] yields similar results when fully trained. Empirically we find the chosen mixture distribution yields faster convergence.

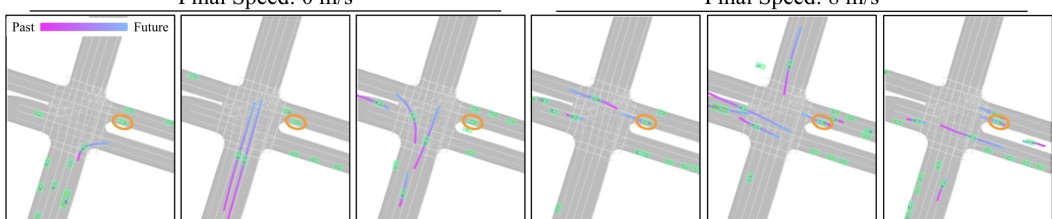

Figure 4: Generated Argoverse scenarios conditioned on the same map image and one agent token with different values for the final speed (2 seconds in the future). The agent corresponding to the agent token is circled in orange. When the final speed is zero the model generates cross traffic agents, and when the final speed is non-zero the horizontal traffic has right of way. In both cases the current pose and speed of the agent token are the same; the only difference is the future speed. Scenario Diffusion is able to infer how the future behavior described in the conditional inputs should impact the placement of additional agents.

Table 3: Agent token speed metrics on Argoverse

| Agent token features | | | | Reconstruction metrics | | |
|:---:|:---:|:---:|:---:|:---:|:---:|:---:|
| Current pose | Extents | Current speed | Final speed | Agent token match rate ($\uparrow$) | Current speed MAE ($\downarrow$) | Final speed MAE ($\downarrow$) |
| $\checkmark$ | $\checkmark$ | | | 0.971 (0.006) | 1.43 (0.17) | 1.57 (0.17) |
| $\checkmark$ | $\checkmark$ | $\checkmark$ | | 0.969 (0.003) | 0.49 (0.01) | 0.79 (0.02) |
| $\checkmark$ | $\checkmark$ | $\checkmark$ | $\checkmark$ | 0.959 (0.007) | 0.49 (0.02) | 0.58 (0.03) |

agent token match rate (what percentage of agent tokens were matched to a corresponding generated agent) and the number of additional generated agents per scene (those unmatched to an agent token). As the value of $p_{\text{mask}}$ increases the agent token match rate stays at 96-97% while the average number of additional agents increases roughly linearly, showing that the model is able to follow both agent and global scene tokens.

### 4.3.2 Token Controllability

To evaluate how well the model uses other information in the agent tokens we compare models that use different agent token features. We run inference conditioned on agent tokens computed from the ground truth sample as in Sec. 4.3.1. For these experiments we use no token dropout (i.e. $p_{\text{mask}} = 0$). We perform the same matching between generated and ground truth agents and report the agent token match rate as in the previous section. For each matched pair, we compute the error of the current speed and final (i.e. $t = 2$) speed. In Tab. 3 we report the mean absolute error (MAE) over all matched pairs, again showing the mean and standard deviation of this metric across multiple models trained with different initial random seeds. As described in Appendix C.3, the speed features are discretized into bins of width 2 m/s, so these models are not expected to achieve perfect reconstruction.

Adding current speed to the agent tokens significantly decreases the MAE for current speed and somewhat decreases it for final speed (as the current and final speed are correlated). Adding a final speed feature to the agent tokens further reduces the MAE for final speed. Adding these additional agent features does not significantly impact the agent token match rate.

### 4.3.3 Relationship Between Agents

One of the key contributions of our work is to generate bounding boxes and trajectories simultaneously for all agents. Doing so allows the generated outputs to capture not just a static snapshot of the scene but also the behavior of various agents. This joint inference of the placement and behavior of agents is particularly important for controlled scenario generation using agent tokens. The behavior described by agent tokens influences where other agents can be placed. Approaches that separate initial placement and behavior into two models are not able to perform this joint reasoning.

Fig. 4 shows scenarios generated using one agent token, with features for the position, heading, extents, current speed, and final speed (i.e. at +2s). Both cases use the same position, heading, extents, and current speed (set to 0 m/s); the only difference is the final speed. For this vehicle stopped at an intersection, its future behavior implies which lanes have right-of-way through the intersection. When

| Evaluated on: | SEA Scene | | | SF Scene | | |
| Trained on: | LV Model | SEA Model | SF Model | LV Model | SEA Model | SF Model |

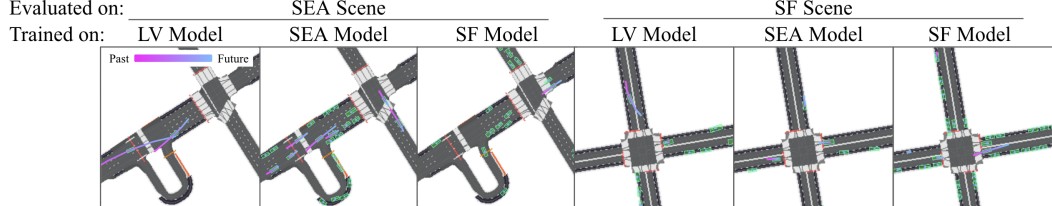

Figure 5: Comparison of scenarios for the internal dataset generated by models trained on only one urban region. While best results are obtained by using a model trained on data from the region of interest, many scenarios generated by models trained in other regions are plausible for validation.

| Evaluated on: | LV Scene | | | SLAC Scene | | |
| Trained on: | LV Model | SLAC Model | Full Model | LV Model | SLAC Model | Full Model |

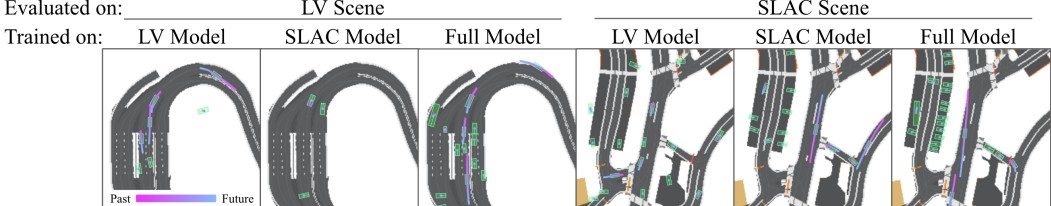

Figure 6: Comparison of scenarios for the internal dataset generated by models trained on data from distinct regions. SLAC is a suburban research campus, notably different from the other three regions. The model trained only on SLAC produces notably worse results when applied to other regions, and the model trained without data from SLAC fails to pick up on unique aspects of the campus road network. The model trained on all regions is able to generate high quality scenarios in both regions.

the future speed is 0 m/s (resp., 8 m/s) the model generates additional agents traveling vertically (resp., horizontally).

## 4.4 Generalization Across Geographic Region

To evaluate the ability to generalize across regions, we train models on data from each region of the internal dataset separately. We then evaluate these models and models trained on the full dataset on validation splits for each region. Results are shown in Tab. 4. Each entry is an average across 5 experiments with different random seeds, with the standard deviation reported in parentheses.

In each region, models trained on that region tend to outperform models trained on other regions. The models trained on the full dataset come close to the region-specialized models. This suggests that there are unique aspects of each region, and that the full model has sufficient capacity to capture this diversity. SLAC is the most distinct from the other regions and shows the largest generalization gap. Additional metrics can be found in Appendix D.3; they follow the same pattern as in Tab. 4.

Fig. 5 shows examples of models generalizing across the three metropolitan regions (LV, SEA, and SF). The model trained on San Francisco data is able to produce reasonable scenes in Seattle, and vice versa. This suggests the potential to produce validation data for new regions before large volumes of driving logs in the new regions have been collected.

Generalizing to dissimilar regions is significantly harder. Fig. 6 shows examples of generalization between LV (a major metropolitan area) and SLAC (a suburban academic campus). Not surprisingly, models trained on each of these two regions performs worse when applied to the other. For example, the LV model does not generate realistic parked vehicles in the parking lots from SLAC and the

Table 4: Generalization across regions

| | MMD$^2$ Positions ($\downarrow$) | | | | Traj On Drivable ($\uparrow$) | | | |
| Model | LV Scenes | SEA Scenes | SF Scenes | SLAC Scenes | LV Scenes | SEA Scenes | SF Scenes | SLAC Scenes |
|---|---|---|---|---|---|---|---|---|
| LV | **0.077 (0.004)** | 0.139 (0.013) | 0.126 (0.013) | 0.204 (0.028) | **0.984 (0.001)** | 0.959 (0.004) | 0.961 (0.005) | 0.917 (0.009) |
| SEA | 0.108 (0.007) | **0.071 (0.004)** | 0.080 (0.005) | 0.141 (0.007) | 0.965 (0.004) | 0.972 (0.002) | 0.976 (0.004) | 0.928 (0.010) |
| SF | 0.110 (0.006) | 0.102 (0.006) | **0.049 (0.002)** | 0.141 (0.008) | 0.978 (0.006) | **0.978 (0.005)** | **0.991 (0.001)** | 0.965 (0.005) |
| SLAC | 0.209 (0.025) | 0.225 (0.020) | 0.193 (0.019) | **0.053 (0.002)** | 0.893 (0.031) | 0.936 (0.017) | 0.950 (0.011) | **0.969 (0.001)** |
| Full | 0.083 (0.006) | 0.084 (0.013) | 0.055 (0.008) | 0.060 (0.009) | 0.983 (0.003) | 0.971 (0.006) | 0.989 (0.003) | 0.963 (0.003) |

SLAC model generates vehicles facing against the lane direction at locations in LV. Fortunately, the model trained on the full dataset is able to produce high quality scenarios in both regions.

## 5 Related Works

A number of prior works have used heuristics or fixed grammars to generate driving scenes [5, 7, 16, 17]. Such approaches are limited in their ability to generate complex and diverse driving scenes. SceneGen [32] and TrafficGen [6] use an autoregressive approach, adding agents one at a time. TrafficGen adds a second stage where a motion forecasting model [33] is used to generate trajectories for the boxes. SimNet [2] uses a conditional generative adversarial network [15] to generate a BEV occupancy image. Heuristic post-processing is applied to identify connected components and fit bounding boxes to them. A second agent-centric model is then used to generate trajectories for these agents over time. In contrast, our method directly produces bounding boxes and trajectories simultaneously for all agents in an end-to-end differentiable architecture. Our approach also does not constrain vehicles to be in lane, as in TrafficGen and several heuristic approaches.

A number of works generate future trajectories for already-existing agents in simulation [14, 31, 37, 39, 40]. Some focus on specifically producing adversarial future trajectories given agents' initial states [4, 10, 35]. These approaches are complementary to our work, and can be used to extend trajectories beyond what is output by Scenario Diffusion.

## 6 Conclusion

In this paper, we demonstrate a novel technique for using diffusion to learn to generate scenarios of dynamic agents moving through an environment for the purposes of testing an autonomous vehicle's ability to navigate through that environment and plan according to those agents. We have shown that our technique leads to models that not only appropriately capture the desired distributions of scenarios and agent trajectories, but allow scenario generation to be controlled to target specific types of scenarios. The fact that diffusion allows declarative information such as tokens to be combined with models learned from data creates the potential for new models that combine other forms of symbolic knowledge such as traffic rules, physical constraints, and common sense reasoning.

**Limitations:** More work is required to show broader generalization. The training data assumes a specific model of perception in the form of bounding boxes and trajectory models. Additionally, we restricted the agent models to on-road vehicle models. While we do not anticipate significant challenges in applying this model to other forms of agents such as pedestrians, we have not incorporated those agents in the research described here. In simulation this approach may need to be extended to iteratively generate agents over a larger region as the AV navigates through the environment.

**Broader Impact:** This paper focuses on developing models to improve self-driving car technologies. There are many positive and negative aspects to the development of self-driving cars that depend as much on the system-wide design and regulatory aspects as these aspects depend on the technical capabilities. However, the focus on simulation in this paper should partially reduce the risks of deploying self-driving cars by providing more effective and systematic coverage of testing scenarios.

**Acknowledgements:** We would like to thank Gary Linscott, Jake Ware, and Yan Chang for helpful feedback on the paper; Allan Zelener and Chris Song for discussions on object detection; Peter Schleede for discussions on diffusion. We also thank the NeurIPS anonymous reviewers, area chair, and program chairs.

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

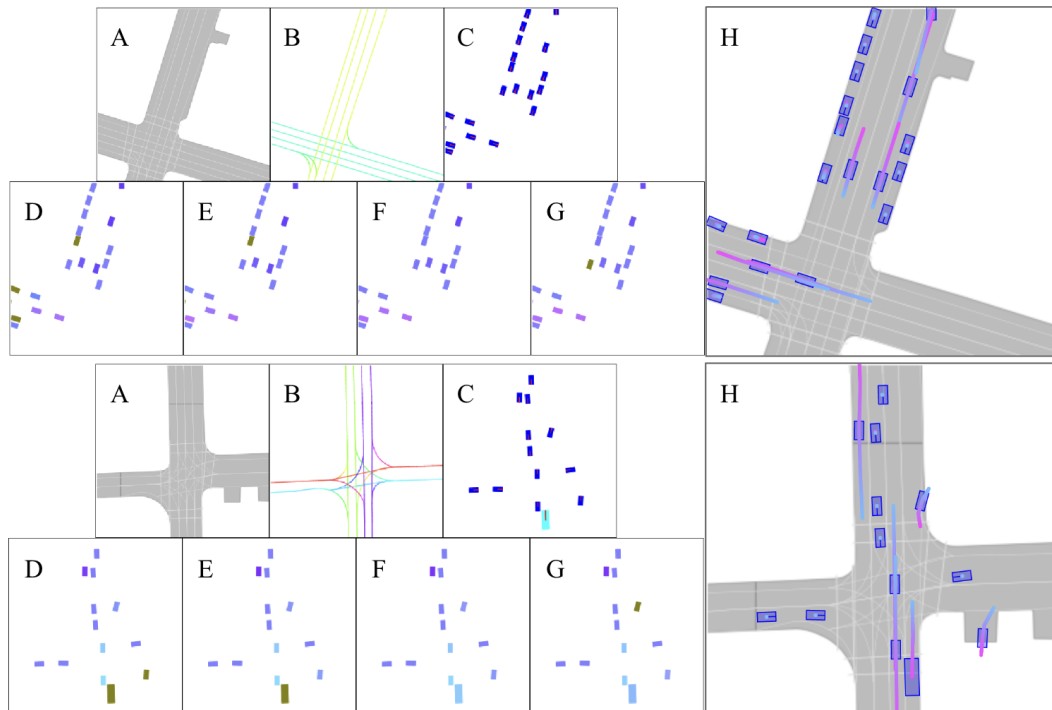

Figure 7: Two examples from Argoverse 2. Images A and B are concatenated in the channel dimension to obtain the 6 channel map image $m$. Image C shows the current location of entity bounding boxes colored by object type with a line to indicate box orientation. Images D-G show the time delta renderings $\Delta_{-2}, \Delta_{-1}, \Delta_1, \Delta_2$, respectively. C-G are concatenated in the channel dimension to obtain the 15 channel entity topdown image $x$. Image H shows the same example using the plotting style used to show generated scenarios.

## A  Datasets

### A.1  Argoverse 2

The Argoverse 2 motion forecasting dataset [36] contains 250,000 driving scenarios, each 11 seconds long. These scenarios cover 6 geographical regions and represent 763 total hours of driving.

To generate inputs for Scenario Diffusion we take all 4s windows (for -2 seconds to +2 seconds) and render an input example for each. To do this rendering we use the provided visualization library to plot a BEV image of the map and add an additional image showing the lane center lines colored by the lane direction. For the entity layers we use the provided visualization library to plot vehicles and buses as rectangles of two different shades of blue. To disambiguate the orientation of the boxes we also render a line from the center of each box to show the heading.

As the argoverse dataset does not include information on agent dimensions, we use constants for each object type (i.e. vehicle and bus) as in the argoverse visualization library. These constants are also used to define the ground truth boxes used to train the autoencoder.

To represent agents' trajectories in the topdown image, we render images for each time delta along the trajectory. For $t \in \{-2, -1, 0, 1, 2\}$, let $p_t = \left(p_t^{(x)}, p_t^{(y)}\right)$ be the agent's position and $m_t$ a mask for whether the agent was observed at the given timestep. We compute four position deltas and corresponding masks:

$$
\begin{aligned}
\Delta_{-2} &= p_{-1} - p_{-2} & \alpha_{-2} &= m_{-1} \wedge m_{-2} \\
\Delta_{-1} &= p_0 - p_{-1} & \alpha_{-1} &= m_0 \wedge m_{-1} \\
\Delta_1 &= p_1 - p_0 & \alpha_1 &= m_1 \wedge m_0 \\
\Delta_2 &= p_2 - p_1 & \alpha_2 &= m_2 \wedge m_1
\end{aligned}
\tag{5}
$$

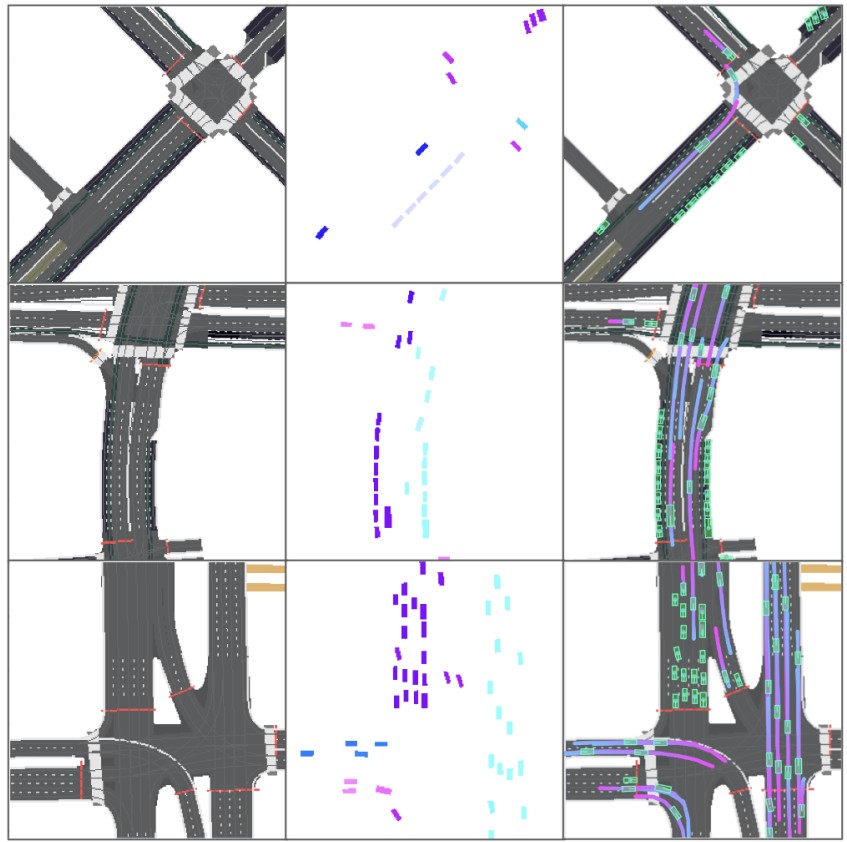

Figure 8: Example topdown inputs for the internal dataset. The left image is the first three channels of the map image $m$. The middle image is the first three channels of the topdown entity image $x$ which fill each bounding box with ones, the cosine of the heading, and the sine of the heading, respectively. The right image shows the corresponding figure using the ground truth boxes and trajectories from the example.

For each $i \in \{-2, -1, 1, 2\}$ we add channels where each agent's bounding box from $t = 0$ is filled with $\alpha_i$, clip $\left(\Delta_i^{(x)}/30 + 0.5, 0, 1\right)$, and clip $\left(\Delta_i^{(y)}/30 + 0.5, 0, 1\right)$.

Scenes are centered on the autonomous vehicle, which is treated as just another vehicle, and cover an area of 100m x 100m. Examples of the topdown inputs $m$ and $x$ for this dataset can be seen in Fig. 7.

## A.2  Internal Dataset

The internal dataset contains 6 million real world driving scenarios from Las Vegas (LV), Seattle (SEA), San Francisco (SF), and the campus of the Stanford Linear Accelerator Center (SLAC). Note that Las Vegas, Seattle, and San Francisco represent dense urban settings, while SLAC is a suburban academic campus. Training and validation data are not separated by location, so a given validation example may have training examples from the same location at other times. Scenarios are sampled every 0.5 seconds from real-world driving logs.

Map information comes from a pre-defined high definition map and vehicle tracks are detected by an on-vehicle perception system. We filter these tracks to only include bounding boxes that overlap with drivable areas in the map (e.g. roads, public parking lots, driveways).

Scenes are centered on the autonomous vehicle, which is treated as just another vehicle during training, and cover an area of 100m x 100m. Examples of scenes from this dataset can be seen in figure Fig. 8.

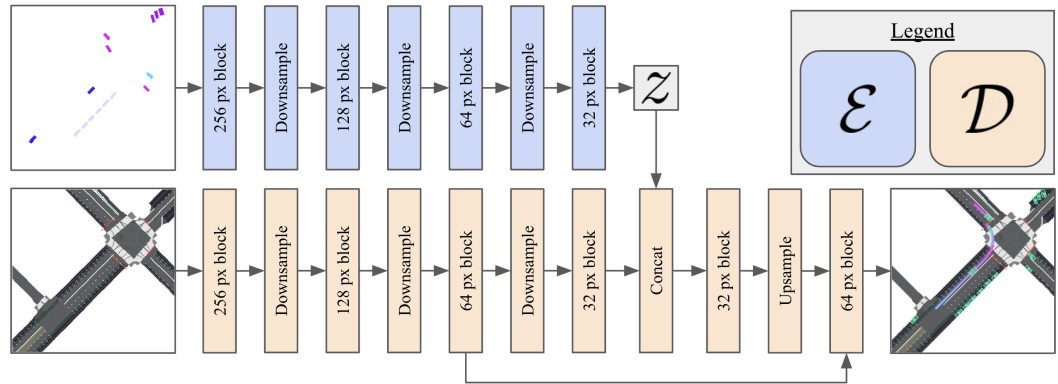

Figure 9: Model architecture for the scenario autoencoder. The architecture used by [24] is modified to include an additional conditional image input (the map image $m$) to the decoder $\mathcal{D}$.

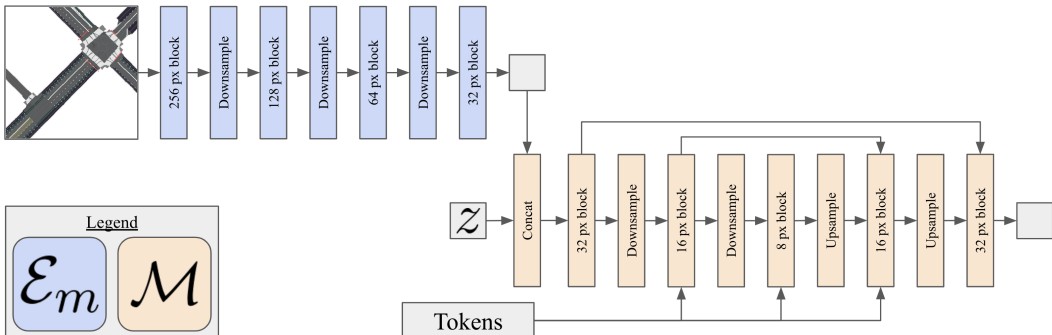

Figure 10: Model architecture for the scenario diffusion model. The denoising Unet $\mathcal{M}$ combines the noisy latent sample $z$, the encoded map information $\mathcal{E}_m(m)$, and the tokens. During training $z$ is generated by running the frozen encoder to obtain $\hat{z} = \mathcal{E}(x)$ and adding noise (see Fig. 2b).

# B   Scenario Autoencoder Details

## B.1   Architecture Details

The architecture for the scenario autoencoder is based on [24] and is shown in Fig. 9. We modify the architecture to add the conditional image $m$ as an input to the decoder $\mathcal{D}(z, m)$. The map image is encoded using the same architecture as $\mathcal{E}$ (but different weights) to obtain a latent embedding with the same pixel dimension. We also truncate the upsampling stage of the decoder to output box proposals at a 64 pixel resolution. The same architecture is used for both the Argoverse and internal datasets.

The 256 pixel and 128 pixel blocks use 64 channels, and the 64 pixel and 32 pixel blocks use 128 channels. We use one Resnet block per resolution.

## B.2   Reconstruction Loss

To compute the reconstruction loss we first match a predicted box to each ground truth box and then apply loss terms between matched pairs. The losses and matching use the same cost functions with different weights.

The classification loss is a binary cross-entropy loss between the predicted box's logit and the probability of the ground truth box (defined to be 1). The coordinate loss applies an L1 loss on the 6 parameters that define the box shape (the cosine and sine of the box heading and the log of the distance to the front, left, back, and right of the box). The vertex loss applies an L2 loss on the positions of the four corners of the box. The trajectory position loss applies an L2 loss on the positions of the trajectory. The trajectory heading loss applies a cosine loss on the headings of the trajectory.

To match a predicted box to a given ground truth box we select the predicted box that minimizes a weighted combination of the classification, coordinate, and vertex losses. The classification term is up-weighted by a factor of 4 over the other two terms.

We apply the five loss functions on the matched pairs. For this loss the classification term is up-weighted by a factor of 20 over the other four terms.

To balance between different motion profiles we categorize each ground truth trajectory as stationary, straight, or turning. To better balance between these three categories we weight the trajectory losses for each category by 0.1, 0.3, and 4, respectively.

We also use a binary cross-entropy loss on the unmatched predicted boxes to have probability 0. As there are many more such negative predictions, we down-weight this loss term. Negative boxes within 3 meters of a ground truth box are down-weighted by a factor of 0.2, and negative boxes more than 3 meters from a ground truth box are down-weighted by a factor of 0.002.

### B.3   Training Details

The scenario autoencoder is trained using the Adam optimizer [19], initial learning rate of 1e-4, and weight decay of 1e-4. The learning rate is decreased by a factor of 10 when the validation loss plateaus. Gradients are clipped to have norm less than or equal to 1. The KL regularization loss $\mathcal{L}_{KL}$ is weighted by 0.1 compared to the reconstruction loss described above. We use 2 RTX A6000 GPUs with a batch size of 16 per GPU.

## C   Scenario Diffusion Details

### C.1   Architecture Details

The architecture for the scenario diffusion model is also based on [24] and shown in Fig. 10. An encoder $\mathcal{E}_m$ with the same architecture as $\mathcal{E}$ with different weights process the conditional map image to the same latent dimension. The noisy latent embedding $z$ and the encoded map information $\mathcal{E}_m(m)$ are concatenated to form the input to the denoising Unet $\mathcal{M}$. In the lower resolution blocks of $\mathcal{M}$, the cross attention layers attend to the conditional tokens. The same architecture is used for both the Argoverse and internal datasets.

The 32 pixel, 16 pixel, and 8 pixel blocks of $\mathcal{M}$ use 64, 128, and 256 channels, respectively. Each resolution block features a single Resnet block. Cross attention is performed with 8 attention heads. The embedded map information $\mathcal{E}_m(m)$ uses 64 channels.

### C.2   Training Details

We use the EDM training algorithm [18]. We set $\sigma_{min} = 0.02$, $\sigma_{max} = 20$, $\sigma_{data} = 0.5$, $\rho = 7$, $P_\mu = -0.5$, and $P_\sigma = 1$. See Table 1 of [18] for more information (note that $P_\mu$ and $P_\sigma$ are referred to as $P_{mean}$ and $P_{std}$ in that table).

The diffusion model is trained with an AdamW optimizer [21], an initial learning rate of 3e-4, and weight decay of 1e-5. The learning rate is decreased by a factor of 10 when the validation loss plateaus. We use 4 RTX A6000 GPUs with a batch size of 64 per GPU.

### C.3   Token Details

In this work we consider the following possible agent token features:

- Current x position relative to the scene center
- Current y position relative to the scene center
- Current heading
- Box length
- Box width
- Current speed

Table 5: Agent token features used for figures.

| Figure | Current Pose | Extents | Current Speed | Final Speed | Heading to Final Position |
|---|---|---|---|---|---|
| Fig. 1 | ✓ | ✓ | ✓ | | ✓ |
| Fig. 3 | ✓ | ✓ | ✓ | | |
| Fig. 4 | ✓ | ✓ | ✓ | ✓ | |
| Fig. 11 | ✓ | ✓ | ✓ | | ✓ |
| Fig. 12 | ✓ | ✓ | ✓ | | ✓ |
| Fig. 13 | ✓ | ✓ | ✓ | ✓ | |
| Fig. 14 | ✓ | ✓ | ✓ | ✓ | |
| Fig. 15 | ✓ | ✓ | ✓ | ✓ | |

- Final speed (at +2s)
- Relative heading from the $t = 0$ position to the $t = 2$ position

A given model will use a subset of these features for its agent tokens. For brevity we use "current pose" to refer to the trio of current x position, current y position, and current heading and "extents" to refer to the pair of box length and box width.

Each agent token feature is processed by a per-feature embedding function; the embeddings of all features are concatenated to get the final token feature vector.

The x and y positions are each embedded using a one-hot vector for 64 bins over the extent of the scene (100m). The current heading is encoded with two channels containing the cosine and sine of the heading. The length and width are encoded with one channel containing the `log1p` of the value. Current and final speed are encoded using a one-hot vector for 10 bins over the interval $[0, 20]$ in m/s. Speeds above 20 m/s are clipped to 20 m/s. The relative heading is encoded using a one-hot vector for 10 bins over the interval $[-0.7, 0.7]$.

Tab. 5 lists the agent token features used for the various figures shown in this paper.

# D    Additional Experiments

## D.1    Autoencoder Ablation Experiments

One of the key hyperparameters for Scenario Diffusion is the latent embedding dimension, the number of channels in $z$. Tab. 6 shows the results of ablation experiments training diffusion models with different autoencoders using different latent dimensions. We find the latent embedding dimension used for experiments in the main paper (4) performs slightly better on MMD metrics and achieves trajectory metrics slighly closer to those of ground truth logs. However, the difference is not statistically significant given the standard deviation across diffusion models trained with different random initial seeds.

## D.2    Diffusion Ablation Experiments

We begin by measuring the effect of changing the probability threshold used during inference. To do so, we repeat the procedure described in Sec. 4.3.1 of sampling scenarios from the Argoverse validation dataset and computing agent tokens from the ground truth agents in the scene. Using $p_{mask} = 0$, we expect the generated agents to match the ground truth agents. We perform the same matching algorithm described in Sec. 4.3.1. Treating the matching as a binary outcome, we

Table 6: Ablation experiments on the autoencoder latent embedding dimension.

| Latent Embedding Dimension | $\text{MMD}^2$ Positions ($\downarrow$) | $\text{MMD}^2$ Headings ($\downarrow$) | $\text{MMD}^2$ Velocity ($\downarrow$) | Traj On Drivable | Lane Heading Difference |
|---|---|---|---|---|---|
| 2 | 0.096 (0.011) | 0.111 (0.009) | 0.058 (0.007) | 0.886 (0.019) | 0.283 (0.032) |
| **4** | 0.093 (0.006) | 0.108 (0.007) | 0.055 (0.003) | 0.895 (0.008) | 0.271 (0.017) |
| 8 | 0.099 (0.009) | 0.116 (0.010) | 0.059 (0.005) | 0.892 (0.010) | 0.282 (0.018) |
| 16 | 0.098 (0.007) | 0.117 (0.009) | 0.058 (0.004) | 0.884 (0.016) | 0.313 (0.035) |
| Ground Truth Log | | | | 0.900 (0.000) | 0.203 (0.000) |

Table 7: Ablation experiments measuring the effect of changing the box probability threshold.

| Probability Threshold | Precision | Recall | F1 |
|---|---|---|---|
| 0.05 | 0.695 (0.086) | 0.981 (0.005) | 0.811 (0.067) |
| 0.10 | 0.886 (0.013) | 0.981 (0.005) | 0.931 (0.009) |
| 0.15 | 0.906 (0.012) | 0.980 (0.005) | 0.942 (0.009) |
| 0.20 | 0.917 (0.011) | 0.980 (0.005) | 0.948 (0.008) |
| 0.25 | 0.925 (0.011) | 0.980 (0.005) | 0.952 (0.008) |
| 0.30 | 0.932 (0.011) | 0.979 (0.005) | 0.955 (0.008) |
| 0.35 | 0.937 (0.011) | 0.979 (0.006) | 0.958 (0.008) |
| 0.40 | 0.942 (0.011) | 0.979 (0.006) | 0.960 (0.008) |
| 0.45 | 0.947 (0.011) | 0.978 (0.006) | 0.962 (0.008) |
| 0.50 | 0.951 (0.010) | 0.978 (0.006) | 0.964 (0.008) |
| 0.55 | 0.955 (0.010) | 0.977 (0.006) | 0.966 (0.008) |
| 0.60 | 0.959 (0.010) | 0.976 (0.006) | 0.968 (0.008) |
| 0.65 | 0.963 (0.010) | 0.975 (0.006) | 0.969 (0.008) |
| 0.70 | 0.967 (0.010) | 0.974 (0.006) | 0.970 (0.008) |
| 0.75 | 0.971 (0.010) | 0.972 (0.006) | 0.971 (0.007) |
| **0.80** | 0.974 (0.010) | 0.968 (0.006) | 0.971 (0.007) |
| 0.85 | 0.977 (0.010) | 0.963 (0.006) | 0.970 (0.007) |
| 0.90 | 0.979 (0.010) | 0.954 (0.006) | 0.966 (0.007) |
| 0.95 | 0.981 (0.009) | 0.940 (0.006) | 0.960 (0.007) |

Table 8: Ablation experiments on the diffusion architecture. The values used in the main experiments are bolded.

| Ablation Variable | Ablation Value | $MMD^2$ Positions ($\downarrow$) | $MMD^2$ Headings ($\downarrow$) | Num Agents EMD ($\downarrow$) | Traj Off Drivable ($\downarrow$) |
|---|---|---|---|---|---|
| Number of Resnet Blocks | **1** | 0.076 (0.009) | 0.091 (0.007) | 5.0 (2.1) | 0.025 (0.003) |
| | 2 | 0.075 (0.006) | 0.089 (0.006) | 5.0 (1.2) | 0.022 (0.002) |
| | 3 | 0.074 (0.004) | 0.090 (0.005) | 4.1 (0.9) | 0.022 (0.003) |
| $\mathcal{E}_m$ Output Dimension | 8 | 0.072 (0.005) | 0.086 (0.004) | 4.6 (1.2) | 0.024 (0.002) |
| | 16 | 0.071 (0.006) | 0.085 (0.006) | 4.3 (1.4) | 0.023 (0.003) |
| | 32 | 0.076 (0.009) | 0.090 (0.008) | 5.1 (1.9) | 0.023 (0.003) |
| | **64** | 0.073 (0.003) | 0.089 (0.004) | 4.1 (0.8) | 0.026 (0.002) |
| | 128 | 0.077 (0.007) | 0.093 (0.006) | 5.3 (1.6) | 0.024 (0.004) |
| Cross Attention Levels | 32px, 16px, 8px | 0.078 (0.007) | 0.097 (0.005) | 5.3 (1.8) | 0.025 (0.005) |
| | **16px, 8 px** | 0.076 (0.006) | 0.090 (0.005) | 5.1 (1.4) | 0.023 (0.002) |
| | 8 px | 0.078 (0.007) | 0.093 (0.007) | 5.5 (1.5) | 0.025 (0.004) |

can define each matched pair of a ground truth and generated agent to be a "true positive" (TP), ground truth agents that are not matched to generated agents as "false negatives" (FN), and generated agents that are not matched to ground truth agents as "false positives" (FP). From these numbers, we can compute precision $= TP/(TP + FP)$, recall $= TP/(TP + FN)$, and F1 $= 2 \cdot$ precision $\cdot$ recall$/$ (precision $+$ recall). Note that this definition of "recall" is the "Agent token match rate" and "false positives" are the "additional generated agents" from Tab. 2. Precision makes sense to measure in this context because we fix $p_{mask} = 0$, so the number of additional generated agents should be close to 0.

These results are shown in Tab. 7, reporting the mean and standard deviation across multiple models trained with different random initial seeds. We observe that precision increases and recall decreases as the probability threshold increases. The mean F1 score is maximized with a probability threshold of 75% or 80%. From this, we select 80% as the threshold to use for most experiments. However, we note that the differences in F1 score are relatively small for probability thresholds between 40% and 95%.

Tab. 8 shows the results of ablation experiments on several key hyperparameters of the diffusion model architecture using the internal dataset. The values used in the experiments for the main paper are bolded. In general, we find that increasing the model size does not significantly improve the available metrics.

### D.3 Cross Region Generalization

Tab. 9 shows additional metrics from the evaluation described in Sec. 4.4. These metrics follow the same pattern as Tab. 4. This table introduces a new metric: the earth movers' distance (EMD) on the number of generated agents. This metric measures how much the probability mass needs to move to

Table 9: Generalization across regions

| Model | MMD$^2$ Heading (↓) | | | | EMD for Number of Agents (↓) | | | |
|---|---|---|---|---|---|---|---|---|
| | LV Scenes | SEA Scenes | SF Scenes | SLAC Scenes | LV Scenes | SEA Scenes | SF Scenes | SLAC Scenes |
| LV | **0.091 (0.003)** | 0.180 (0.008) | 0.186 (0.012) | 0.265 (0.028) | **4.2 (1.2)** | 6.5 (1.4) | 13.3 (1.5) | 12.3 (1.5) |
| SEA | 0.177 (0.011) | **0.075 (0.002)** | 0.137 (0.007) | 0.222 (0.010) | 8.3 (2.1) | 1.9 (0.6) | 8.8 (1.4) | 9.3 (0.9) |
| SF | 0.160 (0.007) | 0.144 (0.004) | **0.048 (0.001)** | 0.197 (0.005) | 9.2 (1.8) | 2.6 (0.9) | **3.1 (0.8)** | 9.1 (0.9) |
| SLAC | 0.375 (0.035) | 0.360 (0.029) | 0.341 (0.023) | **0.073 (0.003)** | 18.0 (0.8) | 11.3 (0.5) | 18.0 (0.6) | **2.8 (0.7)** |
| Full | 0.097 (0.009) | 0.103 (0.023) | 0.060 (0.013) | 0.083 (0.012) | 5.2 (1.8) | **1.6 (1.3)** | 4.4 (1.9) | 3.1 (1.7) |

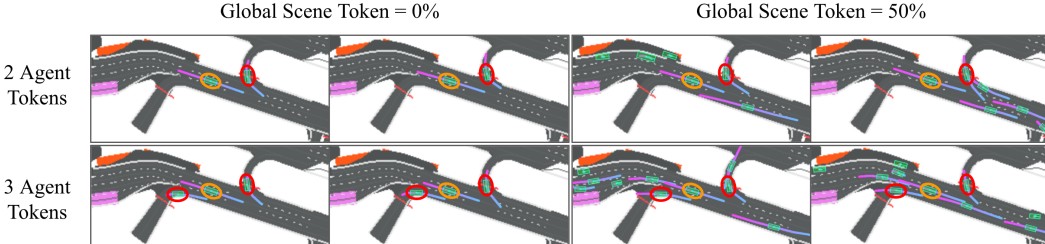

Figure 11: Generated scenarios for the internal dataset. All scenarios are conditioned on two agent tokens; the bottom row of scenarios also uses a third agent token. Agents described by the tokens are circled, with the agent circled in orange representing the AV and other agents circled in red. The model generates a variety of trajectories for the agents described by the tokens. This demonstrates how agent tokens can be used to generate driving scenarios to test a specific category of interactions (e.g. vehicles pulling in front of the AV) while providing diversity within that category.

transform from one probability distribution to another. In this context, the distributions in question are 1D histograms of the number of agents per scene, measured over the ground truth validation scenes and generated scenes using the same map locations.

## D.4 Qualitative Results

Fig. 11 shows scenarios generated using different numbers of agent tokens and different values for the global scene token. Fig. 12 shows adversarial scenarios in a junction created using two agent tokens. These examples illustrate how token controllability allows users to create specific types of scenarios, while letting the model figure out most of the details.

Fig. 13 and Fig. 14 show two more examples of modifying the final speed of agent tokens while keeping all other input features the same. This can be performed with any number of agent tokens.

Fig. 15 takes a ground truth scene from the Argoverse validation dataset, computes agent tokens from two vehicles of interest, and generates novel scenes using these tokens. This process allows users to sample modified versions of a given log to see how the placement of other agents impacts the AV behavior.

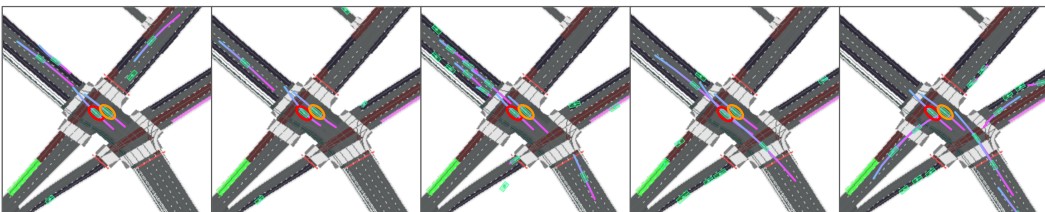

Figure 12: Generated scenarios for the internal dataset showing an agent nudging into the AV's lane while in a junction. All scenarios are conditioned on the same two agent tokens; the corresponding agents are circled, with the agent representing the AV in orange and the other agent in red. In these examples the global scene token is set to 50%. By testing many such scenarios we can find the conditions that result in undesirable driving behavior.

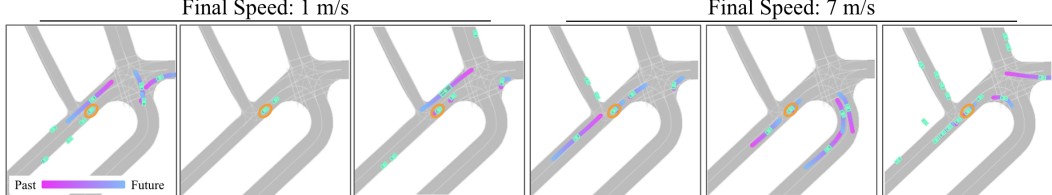

Figure 13: Generated Argoverse scenes conditioned on one agent token, changing only the final speed token feature. The behavior described by the agent token impacts nearby generated agents.

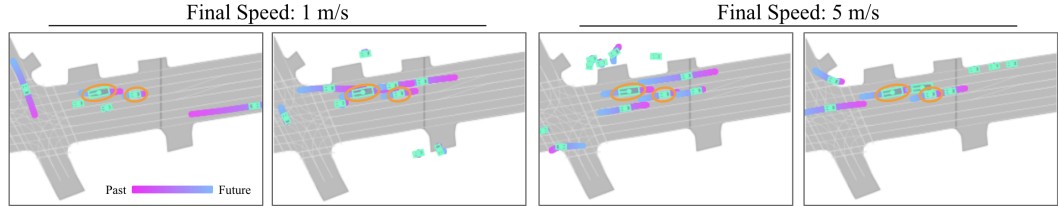

Figure 14: Generated Argoverse scenes conditioned on two agent tokens, changing only their final speed. Both agent tokens are set to use the same final speed. The behavior described by the agent tokens impact nearby generated agents.

## E    TrafficGen

Several modifications were made to TrafficGen [6] accommodate the Argoverse 2 dataset [36]. Argoverse 2 does not contain traffic light information, so such features are not used in the map context. Argoverse 2 also does not include the length and width of agent boxes. In the provided plotting code fixed constants are used for different agent types. We include the "car" and "bus" agent types for training TrafficGen, and update the size prediction head to perform 2-way classification instead of outputting a Gaussian mixture model. To match Scenario Diffusion, we set the model to only generate vehicles in the 100m region around the AV.

Fig. 16 shows scenes generated by our implementation of TrafficGen.

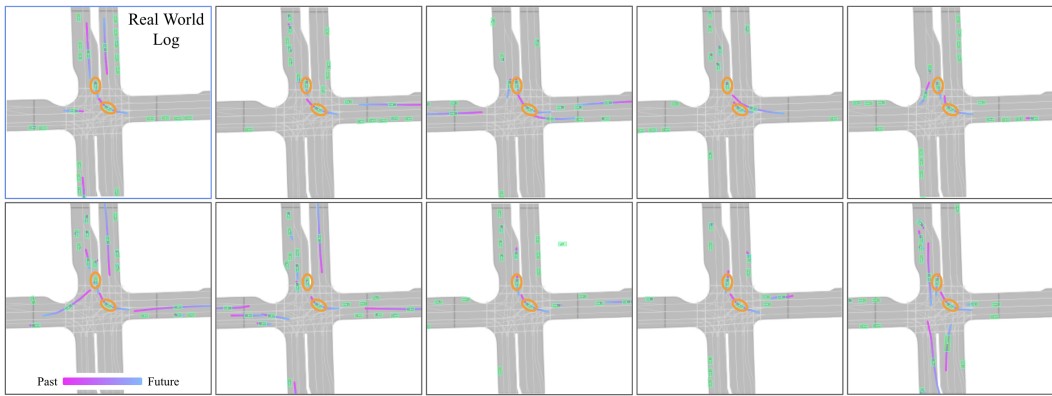

Figure 15: Generated Argoverse scenes using two key agents from a real world log. The top left scene is the original log; the other nine scenes are generated using the same map location and two agent tokens corresponding to the two agents circled in orange. The model is able to produce diverse driving scenes incorporating the two key agents.

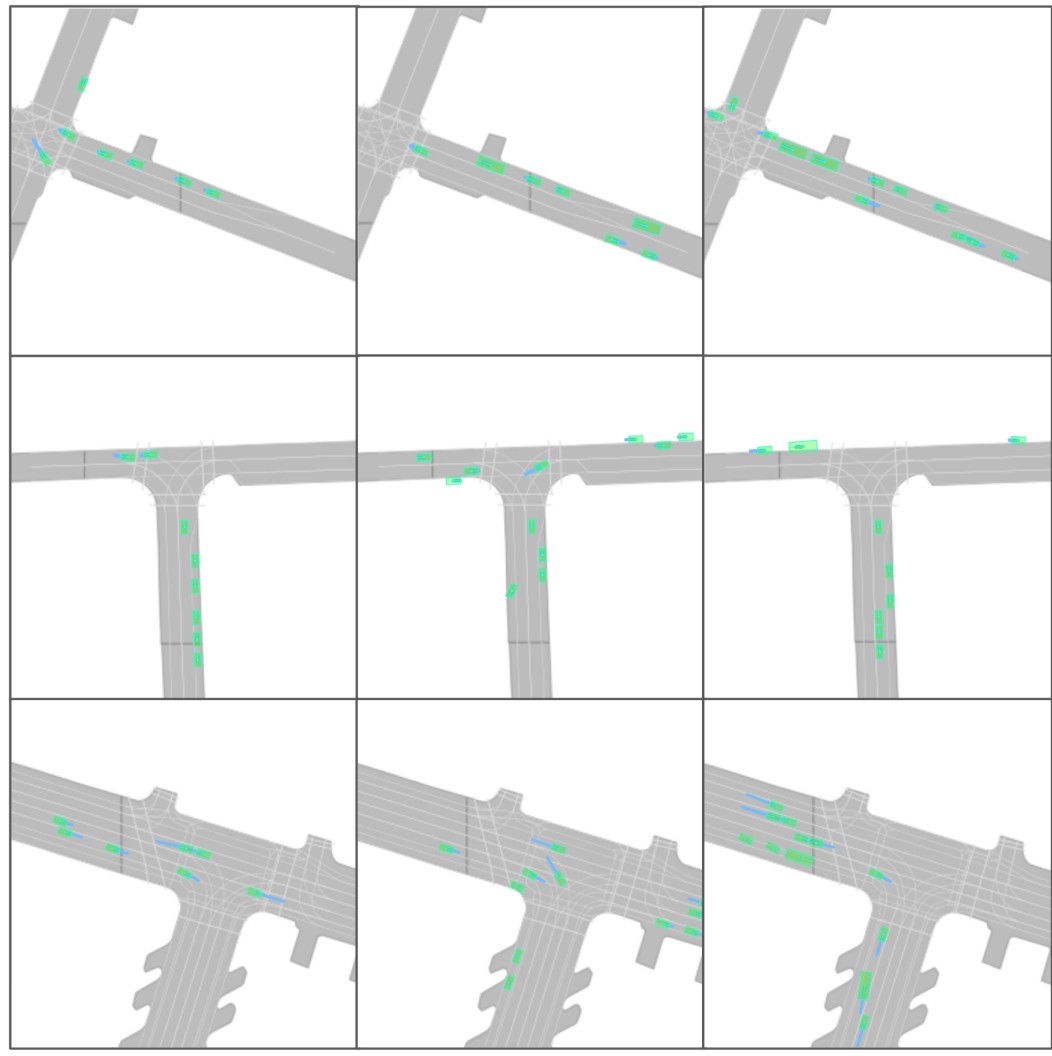

Figure 16: Generated Argoverse scenes from our implementation of TrafficGen. The blue lines show the velocity extended 1 second into the future.

