# OpenReview forum: "Scenario Diffusion: Controllable Driving Scenario Generation With Diffusion"
_NeurIPS.cc/2023/Conference — NeurIPS 2023 poster_

### Official Review · Reviewer_9X4d · 2023-06-21

**Soundness:** 1 poor
**Presentation:** 2 fair
**Contribution:** 2 fair
**Rating:** 4
**Confidence:** 5

**Summary:**

This work adopts a conditional diffusion model to generate the abstract driving scene.

**Strengths:**

It has great potentials to generate future scenes with diffusion model.

**Weaknesses:**

(W1) No comparsion with existing works at all. It is significant to compare with previous methods to demonstrate the superiorty of the proposed method in terms of diversity, generalizability or other aspects. Quantitative comparisons with methods such as SceneGen [1], TrafficGen [2], CTG [3], and Scene Diffusion [4] would help to make the improvement more convincing and easier to recognize. Could the authors provide related results or briefly discuss why these methods were not included in the comparison?

(W2) The proposed method aims to generate all agents simultaneously so that the relationships between them can be learned. Nevertheless, auto-regressive manner also embodies such functionality as it can condition on the generated agents. Have the authors explored the use of auto-regressive ways within the proposed framework?

(W3) As a generation work, a great potential would be producing additional data for training, customizing safety critical scenes for evaluation. However, this part is missing from the current version, which means the claimed benefit is unjustified at all. Could the authors complement some experiments to support that?

[1] SceneGen: Learning to Generate Realistic Traffic Scenes
[2] TrafficGen: Learning to Generate Diverse and Realistic Traffic Scenarios
[3] Guided Conditional Diffusion for Controllable Traffic Simulation
[4] Generating Driving Scenes with Diffusion

**Questions:**

See weakness for details.

I am mostly concerned that this work only decribe their method but do not compare with rich existing literatures at all.  It is hard to tell whether the proposed method really works and is better than existing works.

**Limitations:**

No concern.

---

> ### Author Rebuttal · Authors · 2023-08-09
>
> Thank you for your detailed review!
> * (W1) We have tried to address in the global response why we did not provide a comparison to autoregressive models, and the difficulty of a metric that highlights the need for joint inference, rather than conditional inference. Our hope is that the additional figures in the attached PDF capture this property of our model.
> * (W2) We do not have a substantial and rigorous evaluation of auto-regressive models. Due to the factored nature of auto-regressive models, joint inference of vehicle states would be a challenge for these models.
> * (W3) We agree that experiments showing the applicability of our scenarios in, for instance, control validation, is the long term goal. However, it is not clear how we could add an adequate explanation of such experiments and the results within the space constraints.
>
> Questions:
> * In response to the comment that “it is hard to tell whether the proposed method really works”, we note that our MMD and other metrics are also used by the existing work in the literature such as TrafficGen and we believe are an appropriate basis for at least assessment of competence.

---

> > ### Comment · Reviewer_9X4d · 2023-08-20
> > **Response**
> >
> > Thanks for the reviewers' replies.
> >
> > After discussing with other reviewers, I am still concerned about the experiments. As a new field, it is okay not to compare other works, as they might be concurrent or not fully open sourced. However, consensus is reached that the work should adequately demonstrate that the proposed method is helpful for downstream applications while the current experiments part fail to do so.
> >
> > Thus, my final rating is Borderline reject.

---

### Official Review · Reviewer_VGNk · 2023-06-26

**Soundness:** 3 good
**Presentation:** 3 good
**Contribution:** 3 good
**Rating:** 6
**Confidence:** 4

**Summary:**

This paper proposes a novel approach to use latent diffusion for driving scenario generation. The generated driving scenario is defined as bounding boxes with associated trajectories. The proposed diffusion model is comprised of an autoencoder to encode the BEV image into latent space and a conditional UNet for diffusion modeling. To increase the controllability of the generated scenario, map conditioning and token conditioning are incorporated into the process. The experiments are conducted both on the large-scale internal dataset and the Argoverse 2 dataset, which validates the effectiveness of the proposed algorithm and its flexibility for controllable generation.

**Strengths:**

1. The idea of applying latent diffusion to driving scenario generation is novel and interesting, as diffusion methods have shown great capability in generative tasks.
2. The proposed token conditioning mechanism brings more controllability and the possibility of customization for this task.
4. The overall writing and presentation of the paper are clear and easy to follow.

**Weaknesses:**

1. For the metrics to evaluate the generated scenarios, it is very important to evaluate the validity of the driving scenario, but the paper only measures the rate of off-road trajectories. Other aspects like the rate of collisions between agents, smoothness and plausibility of the trajectories, and possible traffic rules violations should also be measured.
2. Although in Sec 4.3.2 some examples have shown that the proposed method is aware of the relationship between agents, using BEV encoded image to represent the scene without specific designs for agents interactions could not well capture the relationships between agents compared with those GNN-based methods shown in motion-forecasting. Having specific designs to explicitly model the possible interactions is important for the complex driving scenario generation.
4. There is no direct quantitative measurement of how the generated sample faithfully follows the conditioning agent and global token.
3. The goal of scenario generation is to facilitate the development of other autonomous driving tasks like motion forecasting or planning. Therefore, a better way to show the effectiveness and diversity of the generated scenario is by testing them on downstream tasks. Some simple experiments showing the performances of a motion forecasting model training on real and generated data would greatly improve the value of the proposed method.

**Questions:**

1. During the sampling process, would it be helpful to use techniques similar to classifier-free guidance to make the sample ground on the conditioning token more faithfully?
2. Currently the agent token still needs very specific values of different aspects to define its behavior, is it possible to use some more abstract behavior tokens to only specify its overall behavior like overtaking, speed up, right turn, etc. to promote better diversity without much engineering effort?

**Limitations:**

The authors have well discussed the limitations and the potential negative societal impact.

---

> ### Author Rebuttal · Authors · 2023-08-09
>
> Thank you for your detailed review!
>
> We have tried to address the comment about metrics in the global response. We respectfully disagree with the comment that “the paper only measures the rate of off-road trajectories”. We believe that the statistics reported in the different tables comparing the training and generated distributions are appropriate metrics that capture the performance of the model but we appreciate the suggestion for additional metrics.
> * We have tried to address the comment about agent interactions in the global response – we certainly agree that GNN-based methods are a very good way to model agent interactions, but we believe that the diffusion model has shown that it can capture interactions in a scalable manner.
> * We respectfully disagree with the statement that “there is no direct quantitative measurement of how the generated sample faithfully follows the conditioning agent and global token.” We believe that the MMD and EMD statistics exactly show those quantitative metrics, and Table 2 in particular shows the effect of removing the tokens: when every agent has a token, the paper states that the MMD approaches 0, and we see the results when the tokens are masked completely.
> * We agree that showing “the effectiveness and diversity of the generated scenario is by testing them on downstream tasks.” and downstream tasks are the ultimate purpose of our model but have not included these results. Our hope was that our experimental results describing the performance under partial tokenization (Figures 3 and 4, Table 2) and generalization across environments (Figure 5, Table 3) were more informative of the value of our model.
>
> Questions:
> * We very much agree that classifier-free guidance would be helpful, and we have explored using CFG to increase the strength of both map and token conditioning. We have also explored heuristic guidance functions to improve both agent box placement (to avoid overlapping boxes) and to improve trajectories (e.g. to avoid collisions, stay close to lane centers). However, we opted not to include that work in this paper to give more space to describe the essential parts of the approach (scene autoencoder, diffusion, token conditioning).
> * It is definitely possible to use more abstract tokens. We described explicit tokens as they were more obvious demonstrations and were easier to evaluate. The final position heading token feature described in Appendix C.3 is more abstract in that it describes the curvature of the future trajectory but does not directly describe the past trajectory or exact trajectory shape. As seen in Figures 1, 7, and 8 this allows the model to generate different trajectories that fit within the constraints described by the tokens. We have also explored using abstract motion categories (e.g. “lane change”, “turning”, “u-turn”, “stopping”) as token features.

---

> > ### Comment · Reviewer_VGNk · 2023-08-19
> > **Response to the rebuttal**
> >
> > Thanks to the author for further clarifications. After reading the rebuttal and other reviews, my main concern is the evaluation part in terms of diverse and proper metrics for this task, possible baselines, and more ablations.

---

### Official Review · Reviewer_uQ2d · 2023-07-04

**Soundness:** 2 fair
**Presentation:** 3 good
**Contribution:** 3 good
**Rating:** 4
**Confidence:** 4

**Summary:**

This paper tackles the problem of generating both the initial configuration and vehicle trajectories for driving scenarios given an input top-down map. It proposes a latent diffusion model that operates in the latent space of a learned autoencoder trained to decode vehicle boxes and trajectories. The model can be directly conditioned to encourage a subset of agents to adhere to user specifications. Experiments show that once trained on a large real-world driving dataset, the model can produce reasonable scenarios at multiple scales and allows for user control over a subset of agents.

**Strengths:**

The problem is important for AV testing and training.

The proposed method simultaneously generates both an initial scene configuration and vehicle trajectories, instead of in two stages as in prior work.

The proposed latent diffusion model is novel for vehicle trajectories – all prior work I have seen is explicit trajectory diffusion and the design of the autoencoder for such a model is not trivial.

The model allows directly (optionally) conditioning on attributes of one or more agents which gives nice flexibility and allows generating variations of scenarios directly taken from e.g. a log or specified by a user.

Results of generating short 4 sec scenarios look reasonable. Experiments show the ability to generate scenarios of various sizes on the sizable Argoverse 2 dataset. Fig 4 is a neat demo showing that the model is picking up on the semantics of the relationships between vehicles.

The appendix is extensive and provides most of the necessary details to enable reproducibility.

**Weaknesses:**

There are a couple claims that seem over-stated and should be toned down or clarified:
* Conditioning tokens are “more abstract features” (L208) that does not require “fully specifying box and trajectory for the agent” (L206) or “having observed such agents in the real world” (L43) – if I understand correctly, these tokens require a user to give the model the desired agent position, heading, length, width, speed, and sometimes curvature over time (L211),  which is quite a low-level specification of the trajectory and likely will need to be taken from a log. Or do tokens describe the state only at a single $t=0$? If so, this should be made more clear, and I still would not say this is “abstract”.
* The model “generalizes to different geographical regions” (L9) – Table 3 shows seemingly limited ability to generalize across regions with models trained specifically on each region far outperforming other single-region models. The model trained on all regions does reasonably well on each, but this is not generalizing across regions. Moreover, it would be helpful to add a metric that indicates similarity between regions (e.g. MMD between ground truth scenes in A and B) to put the numbers in context. While this experiment is informative about performance across region, it may be a stretch to say it demonstrates generalization across regions.

Some limitations of the technical approach may make the proposed method less practically useful:
* The agent tokens are not hard constraints and in results like Fig 3 the specified agents move around between samples and don’t exactly meet user specifications. A natural solution for diffusion would be to use some kind of test-time guidance as in [37] and [Trace and Pace, Rempe et al., CVPR 2023].
* The 4 sec time horizon (2 sec past + 2 sec future) seems quite short and limits the possible scenarios it can capture.
* The conditioning requires a user to specify all attributes of an agent. But it seems with the masking strategy it should be possible to condition on partial information (e.g. specify the past trajectory but generate the future, or only specifying speed) which would greatly enhance the flexibility.

Some additional evaluation metrics would help to evaluate realism and diversity. For example, MMD on speeds, collision rate between vehicles, and a diversity metric (like average pairwise distance between multiple samples). Additionally, some key experiments are missing:
* The accuracy of the model’s controllability is never evaluated, which is the main claim of the paper. It would be good to report how closely generated agents follow the agent tokens specified by the user (e.g. L2 distance between positions, headings, speed, etc..). The 0% model in Table 2 kind of does this, but the reported MMD metric is not very intuitive.
* A comparison to a baseline to justify the advantages a diffusion model to generate both static states and dynamic trajectories jointly. E.g. a comparison to TrafficGen [6] or a simpler baseline that only generates static states and then uses a trajectory prediction or traffic simulation model to roll out motion, e.g. [BITS: Bi-level Imitation for Traffic Simulation, Xu et al., ICRA 2023]. Or evaluating if the trajectories generated by the diffusion model align with those from a SOTA trajectory forecasting model?
* It would be good to see an example of how these scenarios can be useful. For example, that training a driving policy on a dataset augmented with generated scenarios does better than just a real-world dataset.
* More qualitative results (ideally videos, but additional figures in the supplement would also be helpful).

**Questions:**

Overall, I am borderline on this paper. While the latent diffusion formulation is novel and technically sound and the model nicely generates both starting positions and motion for the scenario, the primary contribution of “controllability” is not sufficiently evaluated, and additional evaluations/baselines would strengthen the paper. In the rebuttal, I would really like the authors to clarify my confusion on the agent tokens and ideally evaluate the controllability of the model by showing how closely it follows user specifications.

The method Sec 3 at times is rather high level and leaves some questions:
* Sec 3.1: why is the encoder not conditioned on the map like in the typical CVAE formulation?
* Sec 3.3: A lot of info about tokens are relegated to experiments in Sec 4.3.1 or the appendix but may be better to include in the methods. I’m still left wondering how exactly the cross attention works, e.g. how are the tokens associated with agents in the scene since there is no canonical agent IDs or ordering? Also, if tokens are specifying the properties of agent trajectories, why do multiple samples of the same scenario (e.g. Figs 1, 7, 8) exhibit varying trajectories for the specified agents?
* How are the number of agents in the scene determined by the diffusion model? Through the agent probability? Why is the global scene token represented as the masking probability and not directly the number of agents that are desired in the scene?
* The methods mentions 2 agent tokens are used, but then 3 are used in Fig 3 and N are used in Table 2, so it’s not clear what is the problem formulation and what’s used in practice.

Another other small suggestion: since explicit trajectory diffusion models are becoming more prevalent [37] [Trace and Pace, Rempe et al., CVPR 2023] [Stochastic trajectory prediction via motion indeterminancy diffusion, Gu et al., CVPR 2022] [Planning with Diffusion for Flexible Behavior Synthesis, Janner et al., ICML 2022], it would be good to discuss them in related work and compare to the proposed latent diffusion.

===================== After Rebuttal ======================

After considering other reviews and discussion with the authors and reviewers, I have decided to keep my rating leaning towards reject.

Given that the problem is relatively new and the latent diffusion approach is novel, I am willing to be lenient on the lack of a baseline comparison. Especially since TrafficGen (ICRA 2023) is concurrent work and SceneGen (CVPR 2021) did not release code. However, without a baseline, the experiments and metrics need to be comprehensive and convincing to show generated scenes are realistic and match user specifications; I don’t believe this is the case in the current draft. The rebuttal Table 1 adds some new map-based metrics, but important realism evaluations like collision rate, plausibility of trajectories (smoothness and comfort), and diversity are still missing. Moreover, the added evaluation of controllability in rebuttal Figure 3 is rather coarse (considered “correct” if within 2.2 m) and does not consider other user specifications like length, width, speed, and curvature.

I also think it would be a strong addition to show results on higher-level semantic conditioning of the model such as vehicle type and maneuver type (e.g. lane change, turning, or u-turn) to justify the claim that the model can handle “abstract” specifications rather than just partial explicit specifications.

**Limitations:**

Limitations are sufficiently discussed in Sec 6.

---

> ### Author Rebuttal · Authors · 2023-08-09
>
> Thank you for your detailed review!
>
> We tried to address the question about additional metrics in the global response, but we agree that we did not provide sufficient detail for the comparison and the advantages of the diffusion model, and will add these to the final paper if accepted. We appreciate the guidance to be more precise about the claims of the paper.
>
> * It is correct that the tokens encode agent position, heading, length, width, and speed, and that these are only initial conditions. However, even with the curvature feature, we do not prescribe the exact path of the trajectory — the model produces variations from the partial description provided by agent tokens. The same architecture can be used with higher level descriptions. For example, we can give position using relatively coarse specifications, and use “bus” vs “car”  tokens instead of the vehicle’s size, and the type of motion (e.g. lane change, turning, or u-turn). We have experiments with these higher level descriptions.
> * We would be grateful for clarification on the comments that our techniques do not generalize. While performance on each training set in Table 3 is definitely better than performance on a different test set, the differences are not significant in every case, e.g., Train/Test on B vs Train-on-B/Test-on-C and we believe this demonstrate some generalization across environments, The largest difference involves environment D, and we are careful in the paper to say that we do not generalize to D, and give specific examples of where and how the models are not generalizing.
> * The agent tokens are not hard constraints by design. We want the model to produce different variations following a partial description via tokens. As you suggest, we have used guidance to refine parts of the generated scenario (e.g. stay close to lane centers, avoid overlapping boxes) but those results were out of the scope of this paper.
> * We used 1-4 seconds of agent history as input and trajectory prediction to be consistent with the literature.
> * We have provided a precision-recall graph of the controllability in the attached PDF, see Figure 3 in the attachment. Our model achieves precision and recall well above 90%.
> * We have tried to address the question about comparison to existing techniques in the global response, but we agree that we did not provide sufficient detail for the comparison and the advantages of the diffusion model, and will add these to the final paper if accepted.
> * We agree that evaluation of the performance of downstream tasks using the generated scenarios is the ultimate purpose of our model but our hope was that our experimental results describing the performance under partial tokenization (Figures 3 and 4, Table 2) and generalization across environments (Figure 5, Table 3) were more informative of the detailed performance of our model.
> We have added additional qualitative figures in the PDF attachment, see Figures 1 and 2 in the PDF.
>
> Questions
> * In Sec 3.1, the map information is provided to the decoder to help the output trajectories be consistent with the map. Since the only outputs of the scene autoencoder are boxes and trajectories we do not need the latent embedding to include information about the map explicitly. Empirically when we tested concatenating the map image to the encoder inputs we saw no change in performance for the scene autoencoder.
> * In Sec 3.3, the cross-attention inside the denoising model is as described in the Latent Diffusion paper. Each pixel from the intermediate representation of the Unet is a query, and the keys/values are agent tokens. When agent tokens include position information the model learns to associate them with queries from the corresponding region of the image.
> * As you correctly point out, there is no fixed agent ID or ordering required. While we do not demonstrate it in this paper, the tokens can be flexible (e.g. multiple tokens describing different aspects of a single agent, a token describing multiple agents, etc) The agent tokens as we define them indicate “there’s an agent in this region with these properties” and the cross-attention updates the latent embeddings in that region to reflect this.
> * Multiple samples have different trajectories because the tokens provide only a partial description of agents. We do not want to exactly dictate the trajectory to allow the model to generate slight variations, hence the term “abstract features”, which we will replace.
> * Regarding the number of agents, if no tokens are used the model learns the distribution of scenes (including number of agents) conditioned on the map. This is measured by the Earth Mover’s Distance metric reported in Table 5. If agent tokens are used, the model still learns the distribution of scenes (including number of agents), but now is conditioned on the map, the number of agent tokens (since every agent token describes a unique agent), and the global scene token which provides information on how many additional agents there are. We experimented with various ways to represent the scene density as a token (e.g. masking probability, total number of agents, total number of agents divided by the drivable surface area) but found no clear advantages in terms of model performance.
> * Regarding the number of tokens, Section 3.3.2 says “we explore two types of tokens”. There can be any number of agent tokens. These tokens are used via cross-attention, which supports an arbitrary number of keys and values. We intentionally demonstrate this capability by showing examples with 2 and 3 agent tokens, and as you point out Table 3 requires using an arbitrary number of tokens. For demonstration purposes we found it better to limit the number of agent tokens to 2-3 so that the model had flexibility to fill in the rest of the scene.

---

> > ### Comment · Reviewer_uQ2d · 2023-08-16
> > **Followup question**
> >
> > Thank you to the authors for the detailed response and clarifications, I really appreciate it.
> >
> > The authors stated that they “have experiments with these higher level descriptions” like “bus” vs “car” tokens instead of the vehicle’s size, and the type of motion (e.g. lane change, turning, or u-turn).” Are these experiments included in the paper or supp? These would be exciting results and showing these examples would greatly strengthen the argument that the model can handle “abstract” specifications. Otherwise, please consider re-phrasing the description to be “partial” specification rather than “abstract”.

---

### Official Review · Reviewer_cvm4 · 2023-07-05

**Soundness:** 3 good
**Presentation:** 2 fair
**Contribution:** 3 good
**Rating:** 4
**Confidence:** 4

**Summary:**

This paper presents a conditional latent diffusion model (LDM) for generating oriented BEV vehicle bounding boxes, each associated with a 4-second trajectory (2s past, 2s future) at a 1s temporal resolution. The autoencoder component of this LDM uses an object detection architecture similar to CenterNet, with rasterized agent inputs. The diffusion model is conditionable on the local map, a global traffic density parameter, and an optional per-agent attribute descriptor. The model is validated on an internal proprietary dataset and Argoverse 2 via a series of ablation studies.

**Strengths:**

What stands out in this work is the overall simplicity and elegance of the framework. It draws on ideas from different communities (generative modeling, object detection, traffic simulation). It also includes interesting original ideas that are potentially applicable to other generative modeling tasks in robotics and self-driving, such as (1) the imbalanced VAE (raster input, vector output); and (2) the masking idea to make agent-level conditioning optional. The writing is clear and mostly easy to follow.

**Weaknesses:**

The key weakness of this work is the lack of any baselines in the experimental analysis. Note that this is not a completely novel task (Section 5 discusses prior techniques). While it is true that not all prior work may be conditionable on per-agent attributes like the proposed model, I believe a comparison to prior work (potentially in the setting without agent token conditioning, as in the last column of Table 2) is warranted to meaningfully understand how well diffusion models perform this task. Secondly, the diffusion model architecture is hard to understand from the main paper in its current form, and a clear description is only available in the supplementary. For example, L_{KL} is marked in Fig. 2a but not described at any point in the main text.

**Questions:**

1. Would it be possible to include one heuristic and one autoregressive approach (as mentioned in Section 5) for the Argoverse dataset as additional columns in Table 2? This is the main limitation of the paper from my perspective, and I am happy to raise my score if it is addressed, or a convincing reason for why such baselines are not useful is provided.
2. Since the conditioning (controlability) is pitched as a key contribution, this should be clearly understandable from Fig. 2b. Would it be possible to expand “M” to show both its encoder and decoder in Fig. 2, with more clarity on how the different inputs/conditions are passed into the diffusion model architecture, instead of a single input arrow?

Minor:

1. I couldn’t understand the caption in Table 3, specifically the text enclosed in brackets
2. [34] and [37] have incorrect citation “year” fields (2022 → 2023)

Update:

Thanks a lot for your clarifications and the efforts made to update the draft.

While I no longer have any issues regarding the method/presentation, I believe the rebuttal does not fully address the concern regarding limited baselines in the analysis of the proposed model. The other reviewers have also commented on the insufficiency of the current experiments to demonstrate a high degree of realism and controllability with the proposed model, and I concur. Therefore, I am maintaining my rating of borderline reject.

**Limitations:**

The key limitations (such as the currently limited generalization capability and simple task definition) are discussed in Section 6.

---

> ### Author Rebuttal · Authors · 2023-08-09
>
> Thank you for your detailed review!
>
> We have tried to address the question about comparison to existing techniques in the global response, but we agree that we did not provide sufficient detail for the comparison and the advantages of the diffusion model, and will add these to the final paper if accepted. We can definitely clarify the diffusion architecture – the $L_{KL}$ regularization loss in Fig. 2a was unfortunately only described in the supplementary material – we apologise for that oversight.
>
> Questions:
> * The suggestion of a comparison to an autoregressive model and a heuristic as additional columns in Table 2 is extremely helpful – thank you for that suggestion. However, we hope that our explanation in the global response provides insight as to why we did not provide the explicit comparison to an autoregressive model in the paper.
> * The suggestion to expand “M” to show both its encoder and decoder in Fig. 2 is also very helpful and we will make that change to the final paper if accepted. It may be helpful to imagine the denoising module “M” being expanded to show something similar to Figure 3 from [Latent Diffusion](http://arxiv.org/abs/2112.10752).
>
> Thank you for the corrections – the text in the parentheses in Table 3 was meant to be an internal comment in the source latex.

---

> > ### Comment · Reviewer_cvm4 · 2023-08-20
> >
> > Thanks a lot for your clarifications and the efforts made to update the draft.
> >
> > While I no longer have any issues regarding the method/presentation, I believe the rebuttal does not fully address the concern regarding limited baselines in the analysis of the proposed model. The other reviewers have also commented on the insufficiency of the current experiments to demonstrate a high degree of realism and controllability with the proposed model, and I concur. Therefore, I am maintaining my rating of borderline reject.

---

### Author Rebuttal · Authors · 2023-08-09

We appreciate the reviewers thoughtful and detailed comments, and agree with the majority of the comments and suggestions. In terms of the overall identified weaknesses, the reviewers’ concerns can be roughly grouped into:
* Absence of comparisons to previous work, especially previous autoregressive models
* Limited quantitative evaluations or metrics.
(along with additional comments from individual reviewers). We believe that these two concerns are linked, and get to the heart of what we were attempting to show in this paper.

One of the challenges of urban driving is that the correlations between the agents are often tightly coupled and cannot be easily factored in scenarios where the agents are forced to interact with each other. For instance, creating a scenario where another vehicle is pulling out in front of our AV cannot easily be generated by an autoregressive conditional model — both our AV and the other vehicle must be placed simultaneously with respect to each other in order to force the interaction. We show our ability to generate this scenario in Figure 1 of our attached PDF.

This is not to say that an autoregressive model cannot be coerced into a limited set of pairwise or joint interactions by regressing on combinations of vehicles, but learning such a model is non-trivial and will not scale as easily as the joint model produced by diffusion. It is worth noting that the previous results such as TrafficGen do not report scenarios that require *specific* forms of interaction — for instance, the interaction results in the TrafficGen paper report interaction metrics as collisions, rather than rate of inter-vehicle responses.

Rather than show this difference experimentally, we attempted to articulate within the text (in the introduction and in section 4.3.2) a principled reason for a diffusion model, and that autoregressive models are by design limited to locally factored representations. We agree that our metrics and evaluations do not necessarily highlight this capability relative to autoregressive models, but it is also not clear how convincing a metric comparison would be. The advantage of diffusion is the ability to learn a high-dimensional joint distribution, capturing many agent correlations as in Figure 4 of our paper. Our best attempt at a metric was the distributional comparisons in tables 1, 2 and 3. We will add additional metrics proposed by the reviewers such as rate of collisions between agents and traffic rules violations, and have added additional metrics in Table 1 of the attached PDF.

We agree that the paper was not clear on the advantages of a joint distribution inferred by diffusion, and that such a model can capture correlations that an autoregressive model does not easily. We will revise the text to better reflect the purpose of diffusion, and show clearer examples.

---

### Decision · Program_Chairs · 2023-09-21

**Decision:**

Accept (poster)

**Comment:**

The paper proposes a driving scenario generator that is controllable using diffusion models. The proposed pipeline combines perception and motion prediction/planning simultaneously. Authors show the approach has sufficient capacity to model diverse patterns and generalizes well to different reigions.

The review score got 4,4,,4 and 6. All reviewers agree that the method / presentation is well stated and easy to follow. The technical novelty is non-trivial, given the problem setting is new and the diffusion approach is novel. The **main concern** is lack of baseline comparison and how to design appropriate evaluation metric.

There are good discussions back and forth among reviewers / authors. AC read the paper carefully, all the rebuttal discussions, and decide to accept the paper nonetheless, based on the following reasons.

- The paper indeed proposes an important probem setup, to generate controllable driving scenarios that can be manipulated with multiple stacks. The proposed method is novel using diffusion. As far as AC's knowledge, there are **few work** on autonomous driving that could utilize diffusion methods to work well. This could **trigger future research** in the community, eg, motion prediction, data augmentation in the sense of foundation model, simulation and scene generation.

- Despite the lack of baselines and evaluation metric is questionable, we should **encourage** novel contributions to be accepted for NeurIPS. Authors did a good rebuttal to address most concerns / technical details, and add addtional experiments to explain.

- The main concern (experimental comparison) raised by reviewers are indeed very important. However, these are fixable and AC trust authors would address them (based on rebuttal and the additional results provided, though not perfect).

As such, it is strongly recommended for authors to revise the paper as sugggested by reviewers (eg. show results on semantic conditioning level by Reviewer uQ2d, etc.) and improve the manuscript to great extent. It is great if the codebase / datasets are open-sourced and results are reproducible as well.